# Sea ice and mixed layer depth influence on nitrate depletion and associated isotopic effects in the Drake Passage – Weddell Sea region, Southern Ocean

Aymeric P. M. Servettaz[1,2,*], Yuta Isaji[1], Chisato Yoshikawa[1], Yanghee Jang[3], Boo-Keun Khim[4,*], Yeongjun Ryu[5], Daniel M. Sigman[5], Nanako O. Ogawa[1], Francisco J. Jiménez-Espejo[1,6], Naohiko Ohkouchi[1]

[1]Biogeochemistry Research Center, Japan Agency for Marine-Earth Science and Technology, Yokosuka, 237-0061 Kanagawa, Japan
[2]Institute of Arctic Climate and Environment Research, Japan Agency for Marine-Earth Science and Technology, Yokosuka, 237-0061 Kanagawa, Japan
[3]Marine Radioactive Monitoring Group, Marine Environment Research Institute, Korea Marine Environment Management Corporation, Busan 49111, Korea
[4]Department of Oceanography and Marine Research Institute, Pusan National University, Busan 46241, Korea
[5]Department of Geosciences, Princeton University, Princeton, New Jersey 08544, United States of America
[6]Instituto Andaluz de Ciencias de la Tierra, Spanish Research Council, 18100 Armilla, Granada, Spain
*Correspondence should be addressed to: A.P.M Servettaz (servettaza@jamstec.go.jp) and B.-K. Khim (bkkhim@pusan.ac.kr)

**Abstract.** The regions near the Antarctic Peninsula in the Southern Ocean are highly productive, with notable phytoplanktonic blooms in the ice-free season. The primary productivity is sustained by the supply of nutrients from convective mixing with nutrient-rich subsurface waters, which promotes rapid phytoplankton growth as the sea ice melts in spring and summer. Surface waters are marked by the contrast between the warmer Drake Passage and the colder Weddell Sea, and seasonal duration of sea ice cover varies accordingly. Sea ice exerts multiple controls over primary production, by shading the light entering the ocean and stratifying the upper ocean with freshening by ice melt. However, the interaction between sea ice and productivity remains poorly characterized because satellites are unable to quantify biomass in partially ice-covered ocean, and direct measurements are too scarce to characterize the seasonally varying productivity. Here we evaluate productivity by assessing removal of nitrate from surface waters by biological nutrient utilization, and study the associated change in $\delta^{15}N$ of nitrate. We use a combination of bottle samples and *in situ* nitrate measurements from published databases, completed by two transects with isotopic measurements. The timing of sea ice melt date conditions the initiation of nitrate drawdown, but the annual minimum of nitrate only weakly correlates with sea ice concentration. As previously reported, we observe that $\delta^{15}N$ of nitrate increases with nitrate depletion. Interestingly, the lowest nitrate depletion and $\delta^{15}N$ values are found in the central region of N-S transects, where intermediate temperature and sea ice conditions prevail. Deeper mixing in waters that passed through the northern Bransfield Strait may explain higher nitrate concentration due to both a greater nitrate concentration at the beginning of the growth season and reduced productivity under light limitation in deeply mixed waters, confirmed by nitrogen isotope modelling. This highlights the importance of oceanographic controls on productivity patterns in sea-iced regions in the Southern Ocean.

## 1. Introduction

In the Southern Ocean (SO), phytoplanktonic blooms develop near the Antarctic continent, in the seasonally ice-covered region and in coastal polynyas (Soppa et al., 2016). The marginal ice zone, where sea ice is present but scattered, contributes to a large part of SO productivity (Arrigo et al., 1998; Savidge et al., 1996). Coastal regions around the Antarctic Peninsula are particularly productive as apparent from satellite-based ocean color scanning (Arrigo et al., 2008; Moreau et al., 2020), which stands out from the low-chlorophyll and high-nutrient waters that characterize most of the SO. The SO is also a major component of oceanic circulation, with upwelling of deep waters contribute to the high nutrient concentration and degassing of carbon into the atmosphere (Marshall and Speer, 2012; Morrison et al., 2015). Understanding productivity patterns and limitations in the seasonally ice-covered SO is thus an important aspect of the global climate.

The Antarctic Peninsula and surrounding waters constitute one of the fastest warming regions in the Southern Hemisphere (Fan et al., 2014; Jones et al., 2016), and yet challenges remain to understand how primary productivity will react to such warming. Although increases in surface water chlorophyll concentration may hint at increasing open-water productivity (Moreau et al., 2015), this method does not account for productivity within the marginal ice zone and may primarily reflect the shift from marginal ice zone productivity to open-water productivity due to longer ice-free season. The presence of sea ice prevents the estimation of chlorophyll concentration from satellites, hampering our ability to evaluate the productivity of seasonally ice-covered regions by remote-sensing methods (Bélanger et al., 2007). While *in situ* quantification of chlorophyll concentration may be useful to estimate biomass of primary producers, it only provides information at a given date, and because such measurements are rarely repeated throughout a growth season, they cannot be used to quantify seasonal productivity and export. Indeed, blooming phases in the SO are marked by a high renewal rate of phytoplankton, with turnover rate of phytoplankton reaching 1 day$^{-1}$ (Arteaga et al., 2020). Alternative productivity estimates have been proposed, relying on the quantification of nutrient uptake by primary producers to estimate production and export in the SO (Lourey and Trull, 2001; Moreau et al., 2020; Nelson et al., 2002; Pondaven et al., 2000). This quantification relies on the seasonal dynamics of nutrient resupply: deep mixing in winter replenishes the nutrient pool, whereas surface stratification during productive season limits exchanges with underlying water, creating a nutrient budget that will be consumed by primary producers (Codispoti et al., 2013). This method does not account for regenerated nutrients within the surface layer by heterotrophic activity during growth season (Fripiat et al., 2015), but is useful to quantify the seasonal export of organic matter as sinking particles (Flynn et al., 2021; Mdutyana et al., 2020).

Nitrate is a major source of nitrogen for microorganisms (Ohkouchi and Takano, 2014), with additional variable contributions of ammonium, nitrite, and urea throughout the season (Goeyens et al., 1995; Mengesha et al., 1998). While $N_2$ fixation can be an alternative nitrogen source in oligotrophic subtropical seas, this contribution is negligible in the SO (Zehr and Capone, 2021). In winter between the ice edge and the Polar Front, nitrification is an important source of regenerated nitrate (Smart et al., 2015), causing the net primary productivity and net nitrogen uptake to be decoupled (Mdutyana et al., 2020). In summer, however, primary productivity is primarily supported by nitrate both in the ice-free SO (Mdutyana et al., 2020) and the seasonally ice-covered zone (DiFiore et al., 2009). In the Weddell Sea (WS), where sea ice concentration (SIC) is usually high, nitrification is slower, suggesting that nitrate utilization may reflect directly net primary productivity (Flynn et al., 2021). Quantifying nitrate depletion in surface water can therefore be used to estimate productivity in summer, or at least organic matter exported to deeper waters. North of the Polar Front in the SO, where there is a strong northward decline in nitrate concentration, there can be significant northward transport of nitrate in surface layer (DiFiore et al., 2006; Sigman et al., 1999). However, this is less of a consideration south of the Polar Front, where horizontal gradients in surface nitrate concentration are typically weak. For the Drake Passage–Weddell Sea region, we aim to evaluate the factors that control nitrate depletion, and thus, net productivity and export.

In the SO, several environmental factors can limit phytoplankton growth and primary productivity. Upwelling of nutrient-rich Circumpolar Deep Water at the Antarctic divergence supplies substantial amounts of nitrate and phosphate, so phytoplanktonic growth may rather be limited by light, iron, silicate, or some combination of them (Franck et al., 2000; Moore et al., 2002). In the Drake Passage and offshore west peninsula shelf break, the dissolved iron content is extremely low even at some locations over the continental shelf ($< 0.1$ nmol kg$^{-1}$), and thus limits productivity (Annett et al., 2017)… The Antarctic Peninsula side of the Weddell Sea receives iron from melting icebergs, which contributes to dissolved iron concentration slightly higher than the central Weddell Sea or the Drake Passage, especially over the continental shelf where concentrations exceed 0.2 nmol kg$^{-1}$ (Klunder et al., 2014). In the coastal regions around the Antarctic Peninsula and nearby islands, dissolved iron is abundantly supplied by desorption from sediment on the shelves or the coasts, glacier melt, and dust deposition (Ardelan et al., 2010; Jiang et al., 2019; Sherrell et al., 2018). Therefore, contrary to the largest part of the SO, iron is not the limiting factor on phytoplanktonic growth in coastal areas such as in the Bransfield Strait (hereafter abbreviated BS, Frants et al., 2013; Measures et al., 2013). Rather, light availability, controlled by ice shading and vertical mixing, limits phytoplankton growth in the BS (Gonçalves-Araujo et al., 2015), which is especially true for winter when light intensity is low (Hatta et al., 2013). This makes it an area of particular interest for studying the relationship between sea ice cover and primary productivity.

Sea ice has two opposite effects on light availability: Ice shades underlying water when it is present, but its melting
releases buoyant freshwater that stabilizes the density structure of upper water column and maintains phytoplankton
community in euphotic zone (Taylor et al., 2013). We thus hypothesize that sea ice melt contributes to thinning the
mixed layer, above which the water is actively mixed by winds. Besides, in areas of sea ice formation, brine rejection
favors winter convective mixing and nutrient influx from nitrate-rich subsurface waters. This replenishes surface water
nutrients, which may then be used next growing season by primary producers. While some algae also develop within
brines and pools of sea ice, their contribution to seasonal biomass productivity is relatively small (Arrigo, 2017). Due
to its complexity and opposite effects, the influence of sea ice on seasonally integrated phytoplankton productivity
remains poorly characterized.
In this study, we explore the relationship between nitrate concentration and sea ice characteristics in the SO near the
tip of the Antarctic Peninsula, to clarify the impact of sea ice on phytoplankton productivity. We first compare satellite-
derived estimations of SIC by area, available year-round, to numerous nutrient concentration that has been widely
measured in the SO both with regular sampling (Olsen et al., 2016) and automated *in situ* quantification (Johnson et
al., 2017). Isotopes can be used to track environmental processes. They are used, for example, in paleoenvironment
studies to infer changes that occurred in the past (e.g. using N isotopes: Studer et al., 2015). Knowing how nitrogen
isotopes relate to nitrate concentration in the modern ocean opens the use of nitrogen isotopes as a tracer of past nitrate
changes. Therefore, we study the link between nitrogen isotopes and nitrate concentration using both observational
data and isotope-enabled simulations. To this end, we describe in further detail a previously unpublished transect of
the concentration and nitrogen isotopic composition of nitrate, interpreted with the help of model simulations in three
oceanic locations.
**2. Oceanographic setting**

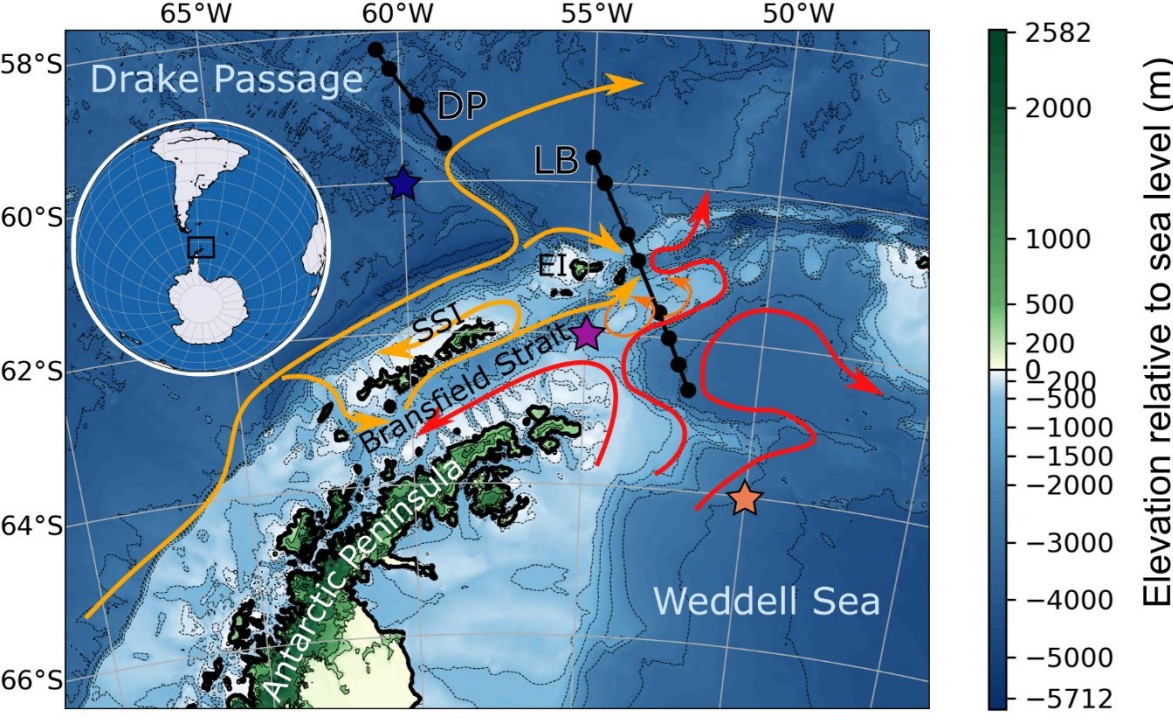

**Figure 1. Regional map of the northern Antarctic Peninsula, Bransfield Strait, South Shetland Islands (SSI), Elephant**
**Island (EI), and parts of the Weddell Sea and Drake Passage. Arrows highlight the surface water circulation, orange for**
**water originating from the west side, and red for water exiting the Weddell Gyre (Moffat and Meredith, 2018; Thompson**
**et al., 2009). DP and LB refer to two transects discussed in Sect. 4.1; black dots represent the location of bottle sampling**
**stations. Elevation from ETOPO 2022 (NOAA, 2022). Colored stars indicate the location of the three points represented in**
**model simulations (Sect. 4.2.4)**
While some studies have described the relationship between sea ice and phytoplankton development in other regions
of the SO (e.g. von Berg et al., 2020; Briggs et al., 2018; Taylor et al., 2013), we focus here on the region around the
northern tip of the Antarctic Peninsula, from DP to WS, with a particular focus on the BS. West Antarctic Peninsula
and BS are coastal regions with non-depleted surface iron (Jiang et al., 2019), meaning that phytoplankton growth is
more susceptible to be limited by light, as modulated by shading and stratification processes related to ice cover and
its melt. Moreover, the West Antarctic Peninsula is the region with the fastest decreasing trend of ice concentration
(Jones et al., 2016), raising the question of how productivity will change with decreasing ice cover.
North of the Antarctic Peninsula, two surface water masses with distinct properties converge in the BS (Fig. 1; Sangrà
et al., 2011). Transitional Zonal Water with Weddell Sea influence (TWW) enters BS from the southeast via the
westward coastal current, running along the tip of the Antarctic Peninsula. In the northwest of the BS, Transitional
Zonal Water with Bellingshausen influence (TBW) enters via a branching derived from the Antarctic Circumpolar
Current, and flows northeastward in the Bransfield Current along the South Shetland Islands (SSI). Further east, this
current divides into a return current north of SSI (Moffat and Meredith, 2018) and an eastward branch passing south
of Elephant Island (Gordon et al., 2000; Thompson et al., 2009). TBW and TWW are separated by a front with a
surface gradient of temperature extending to 50-100 m depth, the Peninsula Front, and subsurface gradient, the
Bransfield Front, located further north beneath the Bransfield Current jet at depths of 150 to 500 m (Sangrà et al.,
2011). The Bransfield Current is associated with TWB-containing anticyclonic eddies in the upper 80 m at its southern
boundary (Thompson et al., 2009). The colder TWW supports the persistence of sea ice for a longer part of the year
in the southeast of the Antarctic Peninsula (Fig. 2). Relative contribution of these two surface water masses to BS, and
the position of the front separating them, has been suspected to vary with westerly wind intensity (Vorrath et al., 2020).
The water masses properties condition the development of primary producers. Previous studies have found higher
concentration of chlorophyll in the TWB, that was interpreted as warm waters with a shallow pycnocline being more
productive (Gonçalves-Araujo et al., 2015; La et al., 2019; Russo et al., 2018).

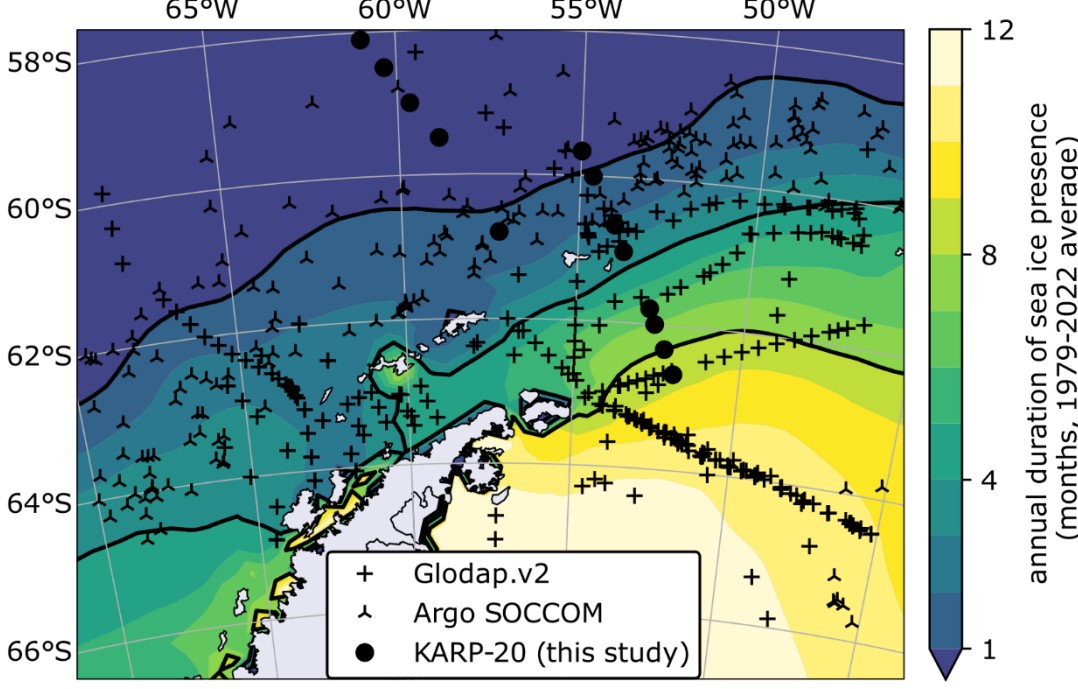

**Figure 2. Map of sea ice seasonality in the Southern Ocean and nitrate data location. Duration of sea ice presence in months**
**per year, defined as the number of months where sea ice concentration is greater than 15% by area (color scale, with black**
**contour lines at 1, 4, and 8 months). Sea ice in coastal areas is reportedly inaccurate due to the coarse resolution of the**
**sensor (Lavergne 2023). Symbols indicate the location of nitrate concentration data used in this study, classified by data**
**source.**

## 3. Material and methods

A large number of measurements have been made publicly available in recent years through data repositories. We briefly describe the datasets used, and additional original data presented in this study. We use a regional subsection of the datasets around the Antarctic Peninsula (Fig. 2).

### 3.1 KARP-20 Nitrate concentrations and isotopes

The Korea Antarctic Research Program 20th expedition (KARP-20) conducted Conductivity-Temperature-Depth (CTD) profile using SBE CTD 911plus along three transects in the DP region between the 1st and the 31st of December 2006. Water bottle samples were collected for nitrate analysis at 13 stations (black circles on Fig. 2) with 12 water depth sampled at each location. Salinity and temperature from CTD were used to determine the potential density anomaly, $\sigma_0$.

Nitrate concentration of each bottle was determined after reduction of nitrate and nitrite to nitric oxide using a V(III) reagent, then quantified by chemiluminescence (Braman and Hendrix, 1989). Our concentration measurements include nitrite, whose contribution is expected to be small in the SO in summer (Thomas et al., 2024). In addition to nitrate concentration, the $\delta^{15}N$ of nitrate was analyzed using the "denitrifier method", where nitrate is converted to nitrous oxide gas by a strain of denitrifying bacteria lacking nitrous oxide reductase activity (Sigman et al., 2001). After transformation of nitrate to nitrous oxide, the nitrogen isotopic composition of resulting nitrous oxide is analyzed with a custom-built in-line preparation and purification system connected to a gas-source isotope ratio mass spectrometer (Thermo Fisher MAT253) at Princeton University (USA) (Weigand et al., 2016).

The denitrifier method analyzes the $\delta^{15}N$ of both nitrate and nitrite. Nitrite can have substantially lower $\delta^{15}N$ due to nitrate-nitrite equilibrium isotope effect (Kemeny et al., 2016). Given this interconversion between nitrate and nitrite N, the isotopic effect of nitrate assimilation affects the $\delta^{15}N$ of both nitrate and nitrite, hence it is more accurate to consider the $\delta^{15}N$ of the sum of nitrate and nitrite when evaluating the assimilation isotope effect (Fripiat et al., 2019). Acidified water samples might have lost most of their nitrite due to volatility of nitric acid, which would bias the $\delta^{15}N$ values towards the $\delta^{15}N$ of nitrate (Fripiat et al., 2019). However, the KARP-20 samples were not acidified prior to isotopic analysis and thus likely retained both nitrate and nitrite.

### 3.2 Glodapv2 nitrate concentrations

The Global Ocean Data Analysis Project Version 2 (GLODAPv2; Olsen et al., 2016) is a collection of biogeochemical bottle data, extending from the World Ocean Circulation Experiment database with quality-controlled additions from more recent cruises. A thorough description of the GLODAPv2.2022 used here is given by Lauvset et al. (2022). Nitrate concentrations in bottle samples are typically determined using the colorimetric absorbance measurement after reduction to nitrite (Armstrong et al., 1967). Measurements included in GLODAPv2 database were quality checked and outliers are removed from the published dataset (Key et al., 2015). Here we only briefly describe how we selected samples used in our study. We defined a regional subset around the Antarctic Peninsula with 45 °W–78 °W and 56 °S–66 °S boundaries, and selected stations where sea ice was present in the year prior to sampling, thus excluding the permanently open ocean zone (see Sect. 3.5 for sea ice data used). Bottles included in this subset were collected between September 1989 and January 2016. In addition to the nitrate value, we use the potential density anomaly ($\sigma_0$) for definition of mixed layer depth (MLD). In the subregion defined for the present study, a total of 187 stations with at least one nitrate measurement above MLD were used (the calculation of MLD used here is given in Section 3.4). While nitrate concentration measurements are very precise with around 0.2% uncertainty, comparison of deep bottles (>1000 m) with nearby (<250 km) measurements give variations usually lower than 2% of the measured value (Aoyama, 2020), which can provide a base estimate of consistency of repeat nitrate measurements, and adjustment were made in GLODAPv2 which resulted in a similar consistency level (Lauvset et al., 2022), even if it is not uncertainty in the strict sense.

### 3.3 SOCCOM Argo floats *in situ* nitrate estimation

Recent developments of ultraviolet spectrophotometers nitrate sensors provide *in situ* quantification of nitrate
concentrations using its absorbance in the ultraviolet band, without requiring chemical transformation (Johnson et al.,
2013; Johnson and Coletti, 2002; MacIntyre et al., 2009). When mounted on profiling floats, they can profile the
nitrate content of the water column, and are set to measure 300 profiles at a rate of one profile every ~5 days which
can make the battery last for a couple of years (Johnson et al., 2013). The SO Carbon and Climate Observations and
Modeling (SOCCOM) has deployed 295 floats in the SO, equipped with biogeochemistry sensors including
submersible nitrate sensor (as of May 2024. source:
https://www3.mbari.org/soccom/tables/SOCCOM_float_performance.html). A description and evaluation of
accuracy of nitrate sensors used on SOCCOM floats is given in Wanninkhof et al. (2016). We use a quality-controlled
dataset where nitrate sensors were calibrated prior to deployment, and offset and drift were adjusted throughout the
float service time using known concentration at depths below 1000 m as a reference for calibration (Maurer et al.,
2021). This lowers the uncertainty on nitrate concentrations to 0.5 µmol kg$^{-1}$. SOCCOM floats have previously been
used to assess nitrate drawdown and net community production (Johnson et al., 2017), and hereafter we specifically
assess the influence of sea ice on nitrate drawdown. We use a total of 194 profiles from 16 floats, all measured in the
regional subset previously defined, dated between January 2016 and June 2022.

**3.4 Definition of mixed layer depth and surface nitrate depletion**

The surface mixed layer depicts a layer actively mixed by wind activity, leading to homogenous physical and chemical
properties. Its depth is controlled by the strength of winds and the density gradients, such as a change of temperature
or an increase of density indicates the MLD. In the SO, it is preferable to use a density criterion due to the low
temperatures throughout the year and significant contribution of meltwater to salinity changes. Therefore, we define
the MLD as the depth with a density increased by $\Delta\sigma = 0.03$ kg m$^{-3}$ relative to the reference density taken at 10-m
depth (de Boyer Montégut et al., 2004).
Surface nitrate concentrations presented in this study are an average of all available measurement points in the mixed
layer (Fig. 3 shows summer nitrate concentrations). For summer samples, we also use nitrate depletion, defined as the
difference in nitrate concentration between the surface mixed layer and a subsurface water referred to as Winter Water
(WW; Moreau et al., 2020; Spira et al., 2024). WW describes water that was last mixed during the previous winter,
when cold surface temperature leads to higher density, thereby increasing the depth of mixing. Similarly to Flynn et
al. (2021), we define the WW layer as the layer between the MLD and the depth of the temperature minimum within
20 to 200 m below the MLD. WW serves as a reference for ocean conditions before the seasonal growth of
phytoplankton, therefore nitrate depletion is defined as the difference between nitrate concentration in the WW
(average of all measurements in the depth range) minus surface nitrate concentration.
Input of meltwater from sea ice with low nitrate concentration could bias this depletion value due to the dilution of
nitrate in surface waters. We corrected for this dilution effect following the method of Flynn et al. (2021), proposed
for the WS. To evaluate the maximum dilution potential, we also use the minimal value for nitrate concentration in
sea ice of 1 µmol kg$^{-1}$ reported for the Bellingshausen and Weddell Seas (Fripiat et al., 2014). Note that, as in the
original study by Flynn et al. (2021), this correction only marginally affects the depletion values; in the samples
considered here, the average summer salinity decrease is 0.18 PSU (1$^{st}$ decile – 9$^{th}$ decile range: 0.04–0.36 PSU), and
result in an average dilution correction of 0.18 µMol kg$^{-1}$ (1$^{st}$ decile – 9$^{th}$ decile range: 0.04–0.37 µmol kg$^{-1}$) for nitrate
depletions. The slightly greater dilution effect for strong salinity decrease is due to the higher seawater nitrate
concentration with high sea ice meltwater contribution. The amplitude of the dilution effect is too small to impact the
conclusions of this article. We hereafter refer to meltwater-dilution corrected nitrate depletion as "nitrate depletion
(corrected)".

**3.5 Sea ice concentration**

We use daily SIC retrieved from version 3 of the EUMETSAT Ocean and Sea Ice Satellite Application Facility sea-
ice products for the 1979-2022 period (OSI SAF 2022a, 2022b; updated from Lavergne et al., 2019). This 25 km
resolution reconstruction is based on microwave emissivity of surface ocean, with SIC calculated from brightness
temperature. Accuracy of SIC by area was evaluated to 8 % in version 2 of this dataset (Lavergne et al. 2019). Updated
processing chain and auxiliary climate fields in version 3 used here result in a reduced bias (Lavergne et al., 2023).
For each nitrate concentration measurement, we extracted the SIC on the grid cell corresponding to measurement
location, at daily resolution on the year preceding the measurement date. We therefore retrieve the sea ice condition
in the time leading up to nitrate measurement. This approach enables monitoring the anomalies in SIC on the year of
measurement, since sea ice conditions at the measurement location may substantially differ from average due to the
high interannual variability (Parkinson and Cavalieri, 2012; Wang and Wu, 2021).
**3.6 Nitrogen isotope modelling**
We simulated the nitrogen cycle in the surface ocean using an isotope enabled ecosystem model. This model has six
compartments, phytoplankton (PHY), zooplankton (ZOO), particulate organic nitrogen (PON), dissolved organic
nitrogen (DON), nitrate ($NO_3^-$), and ammonium ($NH_4^+$). The prognostic variables are the N and $^{15}$N concentrations.
The equations and parameters excluding nitrification are the same as those used by Yoshikawa et al., (2005), which
successfully simulated nitrogen isotope observations in the Sea of Okhotsk, a high-latitude marginal sea. The
equations and parameters for nitrification are the same as those used by Yoshikawa et al. (2022), which includes
photoinhibition terms. The nitrogen isotope fractionation parameters are the same as those used by Yoshikawa et al.
257 (2024).

We applied the model to the ocean environment in three locations around the Antarctic Peninsula: Drake Passage (DP:
60 °S, 60 °W), Eastern Bransfield Strait (BS: 62 °S, 55 °W) and Weddell Sea (WS: 64 °S, 50 °W). This model has
two vertical layers (surface layer: 0-20 m; subsurface layer: 20-120 m). Light intensity at the surface was taken from
long term mean daily net shortwave radiation fluxes of NCEP-NCAR Reanalysis (Kalnay et al., 1996). Water
temperature at upper and lower layers and MLD were taken from the World Ocean Atlas 2018 (Garcia et al., 2019).
Water exchange between the surface and subsurface layers, and between the subsurface layer and the layer deeper
than 120 m, are changed seasonally in conjunction with the MLD. Boundary conditions at 120 m depth for the nitrate
concentration were taken from the World Ocean Atlas 2018 (Garcia et al., 2019) and its $\delta^{15}$N value was fixed to 5‰,
which is in the range of the observations at 120 m during KARP-20 (Sect. 4.2). The model was integrated for a 4-year
spin-up period, and then used to simulate a period of 1 year.
**4. Results and Discussion**
**4.1 Sea ice impact on nitrate depletion**
In this section we assess the impact of sea ice on nitrate depletion, to verify the hypothesis that large seasonal variations
in sea ice could contribute to stratification of the surface water. Sea ice retreat enhances light availability and nutrient
consumption by primary producers (Sallée et al., 2010) while reducing the available nutrient pool due to the thinning
of the surface mixed layer (Smith, and Nelson, 1986; Taylor et al., 2013).
We use the sea ice seasonality at the location where the nitrate concentration was measured. We did not backtrack
either the water parcel or the sea ice, assuming that the seasonality of ice at the point of measurement (fixed location)
resembles that of the water parcel in which nitrate was measured (moving parcel). A parcel-tracking approach would
be technically possible using coupled ocean-sea ice model, but we do not expect significant improvement of our results
due to limitations in modelled ice drift (Uotila et al., 2014).
**4.1.1 Spatial patterns**

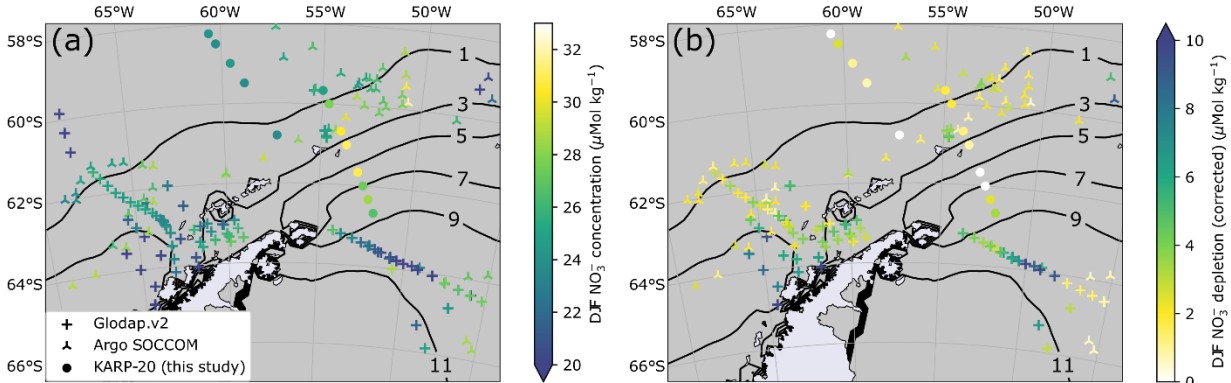

**Figure 3. (a) Map of summer surface water nitrate concentration (color scale) classified by data source (symbol type). Contour lines indicate the 1979-2022 average sea ice presence duration in months per year. (b) same as (a), but for summer nitrate depletion (corrected). The definition of summer depletion and its correction are given in Sect. 3.4. The lower number of points for nitrate depletion results from the absence of nitrate concentration data in the subsurface layer of winter water.**

Spatial patterns of summer (December, January, and February: DJF) nitrate concentration (Fig. 3a) and depletion (Fig. 3b) do not particularly match the sea ice presence gradient (Fig. 2). Nitrate depletion is intense in the western BS and Gerlache Strait, with values consistently exceeding 5 µmol kg[-1], which was attributed to thriving phytoplankton blooms in the very stable surface waters (Castro et al. 2002). In addition, strong depletion values close to 10 µmol kg[-1] are observed in the WS around 50 °W, contrasting with lower depletion values further east, of profiles on the same cruise (Fahrbach, 1993). KARP-20 nitrate concentrations east of Elephant Island (61 °S, 54 °W to 62 °S, 52 °W) are among the highest nitrate concentrations measured for summertime, with values exceeding 30 µmol kg[-1]. KARP-20 stations also reveal that nitrate is not depleted relative to subsurface waters in the outlet of the BS (discussed in further detail in Sect. 4.1). Generally, nitrate concentration or depletion do not correspond spatially to sea ice duration (Fig. 3).

**4.1.2 Timing of ice melt**

The biological activity in the SO is restricted in the austral spring by light, both from low incident light angle and from the shading effect of remaining sea ice. Near the Antarctic Peninsula, the duration of sea ice cover varies latitudinally from the quasi-permanently ice-covered WS to the open ocean in the center of DP (Fig. 2). As the sea ice retreats, the light limitation is expected to be reduced, and the water column is stabilized by release of low-density meltwater (Taylor et al., 2013). This would favor phytoplanktonic growth and nitrate consumption.

To visualize the effect of ice retreat on nutrient drawdown, we show how nitrate depletion (surface minus subsurface concentration, see Sect. 3.4) varies during the season, and relative to the day of ice melt at different locations around the Antarctic Peninsula (Fig. 4). In September and October, the surface nitrate depletion is minimal, owing to deep mixing and limited nitrate utilization. In late spring to early summer (November-December), the surface nitrate is more depleted, although approximately half of the waters sampled during these months have nitrate depletion of less than 3 µmol kg[-1]. In January and February, the surface nitrate is depleted by more than 5 µmol kg[-1] at most sampling stations. Nitrate depletion exceeding 5 µmol kg[-1] is encountered throughout the summer. In March, the deepening of MLD vertically homogenizes the nitrate concentrations, and in conjunction with the slowing of biological uptake in autumn, the nitrate depletion in surface water decreases.

Timing of ice melt controls the initiation of phytoplankton bloom and associated nutrient uptake through the formation of buoyant surface meltwater (Fig. 4). Although few samples were recovered prior to complete ice melt, the nitrate concentrations measured before ice melt are marginally depleted relative to subsurface waters, with less than 5 µmol kg[-1] difference. The highest of these pre-ice melt depletions comes from the profiles measured in July when the water column may not have been homogenized, and the depletion is probably a remnant of the previous year. Low nitrate depletions prior to ice melt results from the low biological activity, the larger nutrient resupply due to well-mixed surface waters, or a combination of both. After sea ice melts, depletion greater than 5 µmol kg[-1] is frequent, but more than half of nitrate depletion values are still lower than 5 µmol kg[-1]. Although there is a large spread of nitrate

depletion values at any time after the ice melt, the highest depletions are reached after about 70 days, which may reflect the entire duration of the phytoplankton bloom following ice melt, with the lowest nutrient values after the bloom phase ends (Arteaga et al., 2020, and supplement therein). During early summer, productivity evolves from regenerated to new production and the nitrate uptake increases as any ammonium remaining from the winter is consumed first (Savoye et al., 2004). Later in the season, phytoplankton growth slows down (Arteaga et al., 2020), and the relative contribution of regenerated nutrients to biological production increases (decrease in the *f*-ratio, Fripiat et al., 2015; Mdutyana et al., 2020; Sambrotto and Mace, 2000). Additionally, weakening of the stratification in later season can lead to decrease in surface nitrate depletion, as mixed layer gradually deepens and incorporates nitrate-rich water from the subsurface. Melting of sea ice can also favor phytoplankton bloom by releasing iron that concentrates in sea ice (Boyd and Ellwood, 2010; Lannuzel et al., 2016), in particular for regions where iron is limiting such as the DP.

In summary, nitrate depletion is generally greatest about 70 days after ice has melted (Fig. 4b). It points out the high nutrient utilization in well-lit buoyant lens of meltwater, supporting the hypothesis of sea ice control on nutrient utilization. However, the large variability of nitrate depletion noted at any point in time indicates that while nutrient utilization may be optimal at a certain timing after ice melts, nutrient depletion will not necessarily occur at this time. Three-season monitoring near Palmer Station suggests that stratification by surface temperature increase after sea ice has melted is the main condition for bloom initiation (Moline and Prézelin, 1996), which can explain the equivocal relationship between sea ice retreat and bloom initiation: temperature remains low as long as sea ice remains, but sea ice melt is not necessarily immediately followed by a temperature increase. Temperature rise may trigger the bloom initiation at a delay with ice melt by enhancing both stratification and productivity.

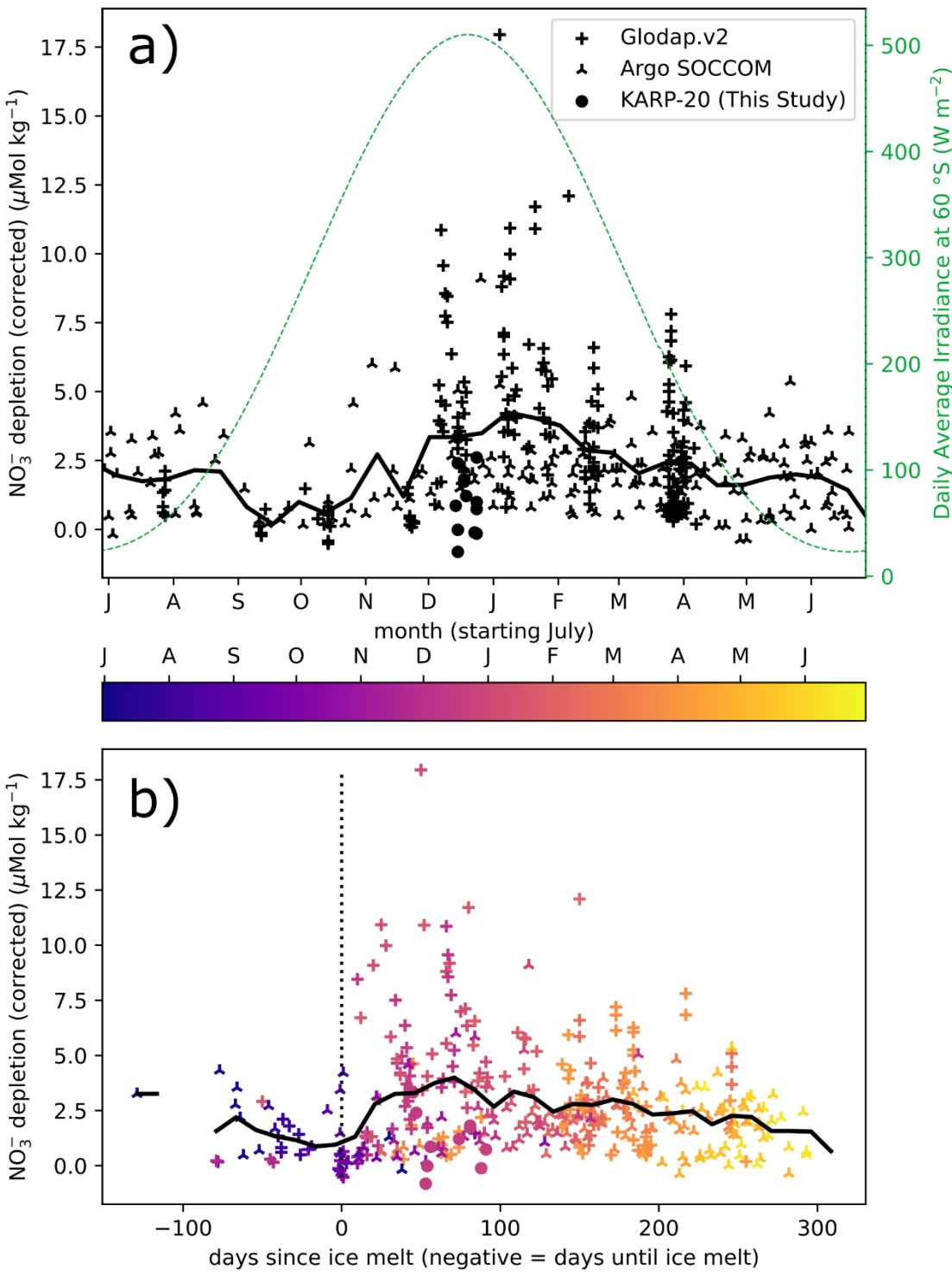

337

**Figure 4. Nitrate depletion as a function of day of year (a) and day since ice melt (b), defined as the number of days spent with ice concentration below 15%. The annual cycle of insolation is given as daily average irradiance at the top of atmosphere at 60 °S (green dashed line). The color of points in (b) indicates the timing in the year. The 25-day rolling average is represented by tick black continuous line on both plots.**

**4.1.3 Seasonal range of ice concentration**

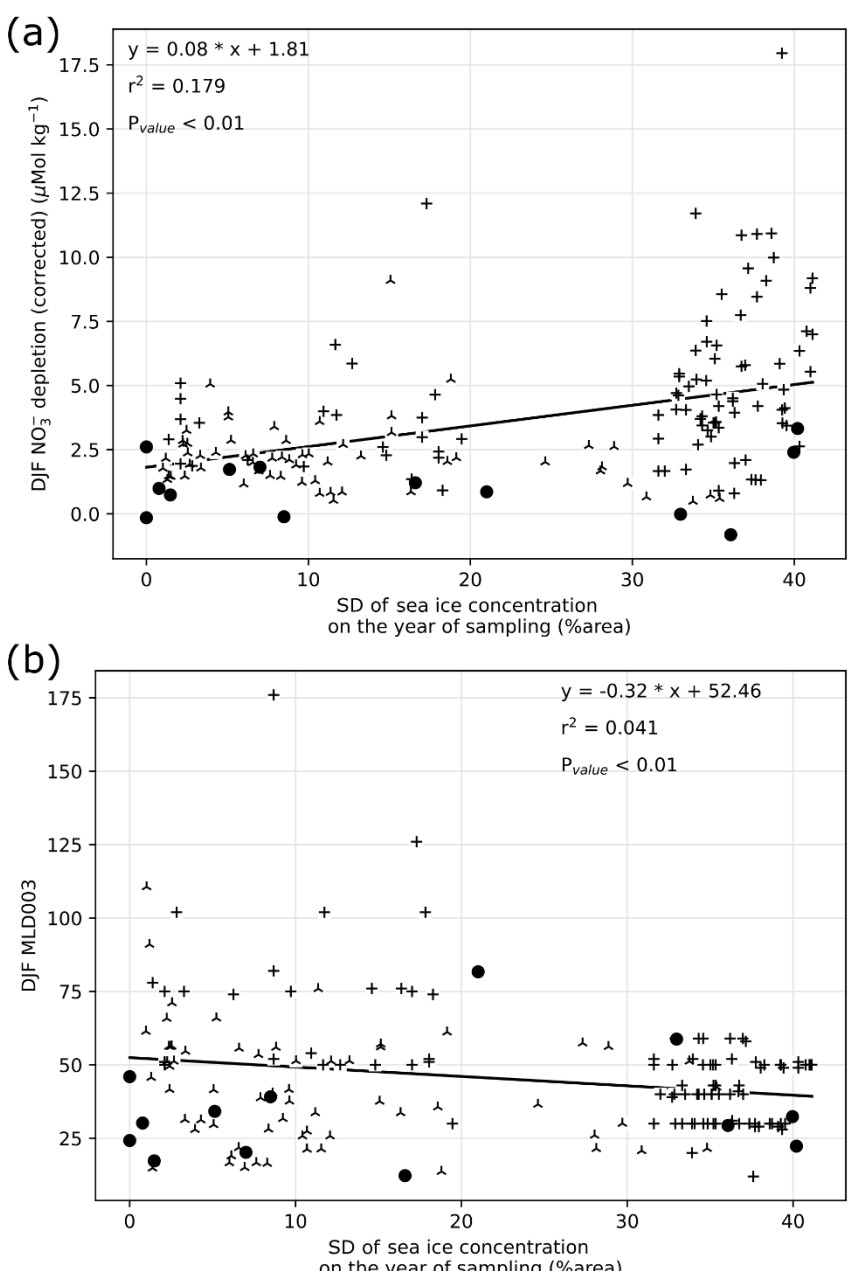

**Figure 5. (a) Summer (December, January, and February: DJF) nitrate depletion as a function of sea ice seasonal range estimated with yearly standard deviation of sea ice concentration. (b) summer (DJF) mixed layer depth (MLD) as a function of sea ice seasonal range. Data source is indicated by symbols, consistently with previous figures.**

In this section we investigate whether an increased seasonal variation of the SIC (calculated as standard deviation of annual sea ice at daily resolution) would favor nitrate depletion, through the shoaling of the MLD which improves the light availability while reducing the amount of nutrients (Fig. 5). Although volume or mass of sea ice would be more suited for this comparison, parameters relying on thickness are poorly constrained in the SO (Kwok and Kacimi, 2018; Williams et al., 2015). We thus rely on satellite data to estimate the area of sea ice coverage. We use standard deviation of SIC (SD$_{sic}$) over the year leading up to the sampling to evaluate the range of seasonal changes in sea ice cover. It is preferred over average sea ice to rule out high sea ice cover grid points such as in the WS, where SIC is high year-round, leading to high average and low variability sea ice cover.

Summer nitrate depletion varies widely at the regional scale studied here, with values ranging from 0 up to 12.5 $\mu$mol kg$^{-1}$. While most nitrate depletion greater than 5 $\mu$mol kg$^{-1}$ occurs at SD$_{sic}$ higher than 10%, there is nevertheless a wide range of nitrate depletion encountered at any SD$_{SIC}$ (Fig. 5a). Notably, non-depleted (depletion < 1 $\mu$mol kg$^{-1}$) waters are observed regardless of the SIC. A weak positive correlation (r$^2$ = 0.179, p$_{value}$ <0.01) between nitrate depletion and SD$_{SIC}$ suggests that about 18 % of the variability of both sea ice and nitrate depletion is shared. Although we cannot conclude on a causal relationship, changes in SIC may at most be responsible for 18% of the variability in nitrate depletion. This low correlation indicates that seasonal amplitude of sea ice alone cannot explain all the variability found in the surface nitrate depletion in summer in this region.

We also tested if high SD$_{sic}$ leads to reduced MLD, through the meltwater supply to the surface (sample locations are mostly ice-free in summer, except for a few locations with partial ice cover). Because we analyze the sea ice seasonality at a fixed location for simplicity, this relies on the assumption that sea ice melts locally, ignoring possible ice drift and/or meltwater horizontal transport. Comparing summer MLD with yearly SIC (Fig. 5b) reveals that while stations with yearly SIC exceeding 20 % have a MLD consistently shallower than 60 m, a wide range of MLD is encountered at stations with low SIC, implying control of MLD by additional factors. Indeed, sea ice melting controls freshwater inputs and salinity, but temperature is also an important factor of stability in the sea ice covered section of the SO (Pellichero et al., 2017). Consequently, despite being significantly correlated (p$_{value}$ < 0.01), MLD and SIC only share 5.7 % variability (r$^2$ = 0.057). At the regional scale, sea ice seasonality has a limited influence on MLD. In turn, the summer nitrate depletion is mostly independent of sea ice cover, although it is less frequently depleted in waters with scarce sea ice cover.

Reasons to why sea ice seasonality does not control nitrate depletion at the regional scale likely emerge from the variety of settings across fronts and basins around the Antarctic Peninsula. Light limitation is often indirectly linked to sea ice: the removal of shading is not sufficient to trigger bloom and nutrient drawdown (Fig. 4). It is possible that light limitation is relieved with stratification of the upper ocean layer, which can occur with a delay after ice melts, with temperature-driven stratification. Indeed, density profile and MLD are more likely to be controlled by temperature rather than salinity in the seawaters north of the Peninsula Front (Gonçalves-Araujo et al., 2015), where low inputs of freshwater are compensated by stronger temperature-driven stratification. On the contrary in the WS and southern half of BS, sea ice provides freshwater lowering salinity, but it also buffers the temperature and maintains a cool surface even in the summer. Possible iron supply from ice melt may also increase productivity without changing MLD (Lannuzel et al., 2016), and may partly explain why nitrate depletion is better correlated to SD$_{SIC}$ than MLD. Regional differences in productivity response to MLD were also found West of the AP (Vernet et al., 2008), and chlorophyll *a* concentration in the BS do not appear to correlate with MLD either (Romanova et al., 2021). Given the spatial variability within the study region, it may be hard to infer the general influence of sea ice on productivity for other regions of the SO, which have different limitations. A regional comparison of sea ice extent and chlorophyll a suggests that temperature and nutricline are important drivers in spatial discrepancies (Behera et al., 2020).

## 4.2 Observations and modelling of [15]N enrichment

While nitrate concentration does not appear clearly linked to sea ice, investigating the isotopic signature in nitrate may reveal further constraints on nitrate uptake. Additionally, how the nitrogen isotopes relate to nitrate concentration and other environmental processes is necessary information to be able to use isotopes as a tracer of these processes. In this section, we focus on the DP and BS transects (location given in Fig. 1), with measurements of nitrate concentration and $\delta^{15}$N. We verify that the relationship between nitrate concentration and isotopes follows a Rayleigh distillation in this region (Sect. 4.2.2). We then validate our interpretations against an isotope-enabled nitrogen cycle model forced by environmental parameters taken from three characteristic sub-regions (Sect. 4.2.3).

### 4.2.1 KARP-20 nitrate concentration and $\delta^{15}$N transects

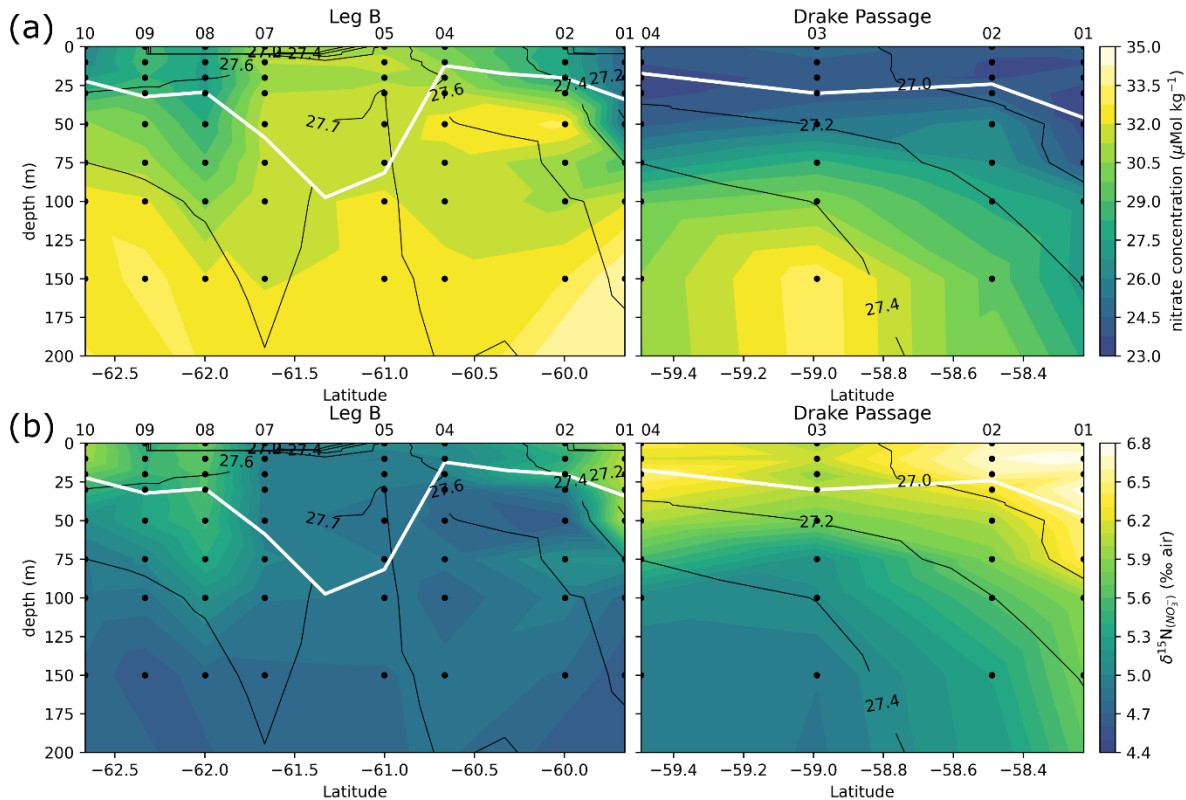

Figure 6. Leg B and Drake Passage Transects, with color shadings of nitrate concentration (a) and $\delta^{15}N$ of nitrate (b). Black contours represent the CTD potential density anomaly ($\sigma_0$ in kg m$^{-3}$). White line highlights the MLD, computed as the depth with a potential density increased by 0.03 kg m$^{-3}$ relative to the density at 10 m depth.

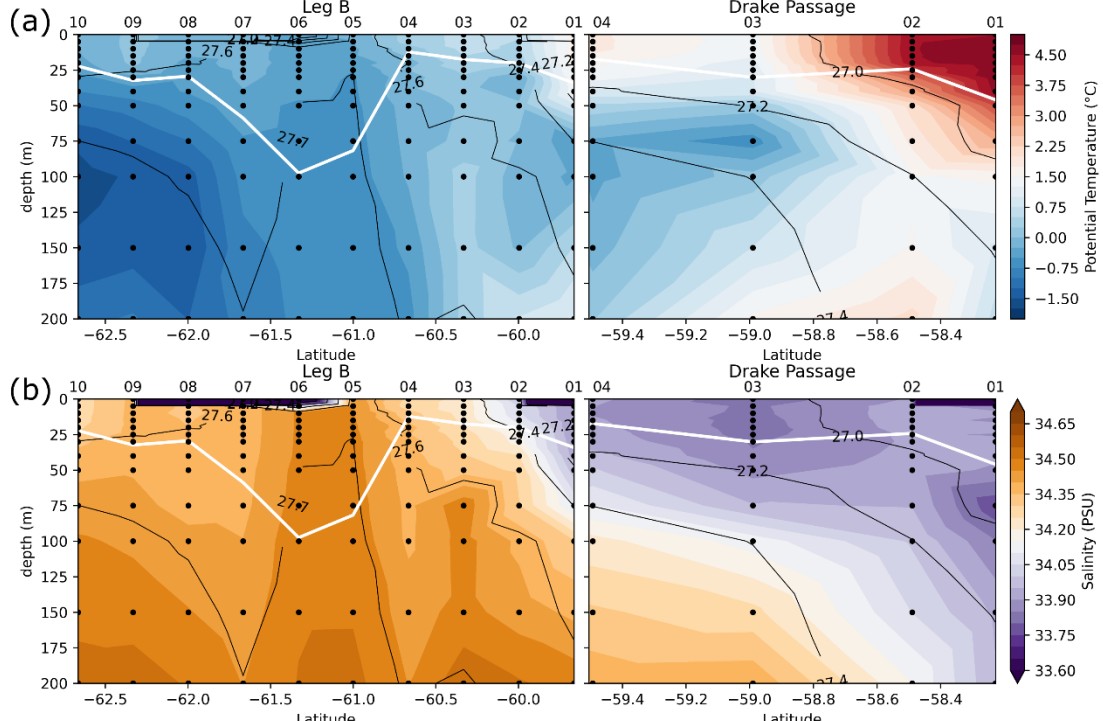

Figure 7. Same as Fig. 6, but for potential temperature (a) and salinity (b).

Transects of nitrate concentration in the DP and eastern BS (Fig. 6a) depict the typical SO summer pattern, with partial
depletion in the surface waters, down to a minimum of 23 μmol kg$^{-1}$ in the DP, compared to subsurface waters in
which concentrations exceed 31 μmol kg$^{-1}$ (profiles also given in Appendix Fig. A1). Isopycnals follow the general
SO pattern, with deepening at lower latitudes. An exception is the center section of LB transect (60.7 °S to 61.7 °S),
where there is a shoaling of the 27.7 kg m$^{-3}$ isopycnal, indicating the presence of high-density water at relatively
shallow depth. The high density is due to higher salinity (>34.45 PSU) of this water rather than temperature (Fig. 7).
This center section also has the highest surface nitrate concentration, about 30 μmol kg$^{-1}$, and the deepest MLD due
to the vertically homogenous density. In the DP, surface nitrate concentration is generally lower, and the nitrate
concentration appears low even beneath the MLD (Fig. 6).
The center section of LB transect with high surface nitrate concentration and deep mixing is in the prolongation of
northern BS current that circulated around the SSI and Elephant Islands (Fig. 1). Although its high salinity could
correspond to Weddell Sea Water transported along the slope on the western side of the Powell Basin, its intermediate
temperature rather supports a western origin (TWB). Physical ocean modelling supported by tidal stations suggests
that tidal interaction on the shelf of the SSI arc actively mixes the water, mixing in high salinity from deeper water
and homogenizing the density profile (Zhou et al., 2020). The vertical homogeneity of density remains when the water
is transported downstream, allowing for the wind mixing to take over and maintain a deep mixed layer, visible in the
60.7–61.7 °S section of the LB transect. Activation energy for wind-mixing is lowered in case of lower density
gradient (Pollard et al., 1973), which means that a similar wind stress will result in a deeper mixing. Surface nitrate
concentration appears closely related to these oceanographic conditions.
Two mechanisms may explain higher nitrate concentration in deeply mixed water: (1) quantitatively greater initial
nitrate pool, due to a larger volume (or greater resupply) meaning that for an equal uptake per area unit, the
concentration remains higher; and (2) lower nitrate uptake due to a limited productivity. Nitrogen isotopes can provide
insight into the relationship between nitrate supply and consumption rates, both of which are seasonally changing.

### 427 4.2.2 $^{15}$N enrichment during Rayleigh distillation

Nitrate uptake by phytoplankton is associated with an increase of δ$^{15}$N of the nitrate remaining in seawater (following
the δ notation relative to the ratio $^{15}$N/$^{14}$N in atmospheric N$_2$), due to the preferential uptake of $^{14}$N by the
microorganisms (Sigman et al., 1999). Consequently, nitrate concentrations are inversely correlated with δ$^{15}$N of
nitrate in the nutrient-rich SO (Lourey et al., 2003).The consumed nitrate is transformed into organic molecules with
an average biosynthetic nitrogen isotope effect ε of around 5‰ (Altabet and Francois, 2001; Sigman et al., 1999;
Waser et al., 1998), given by the difference in kinetics of reaction rates between $^{14}$N and $^{15}$N: $\varepsilon = {}^{14}k/{}^{15}k - 1$.
A system where a component is progressively removed without coincident resupply can be described by the Rayleigh
distillation model. It can be applied to describe isotopic fractionation during nitrate uptake in the condition that there
is net removal from the water through one process (nitrate assimilation), and the removed nitrate has an isotopic
composition that differs from that in the water. In this case, the isotope effect can be quantified by the following
approximation using measurable parameters:
$$\varepsilon = \frac{\delta^{15}N_{initial} - \delta^{15}N}{\ln(NO_3^-) - \ln(NO_{3\ initial}^-)} \tag{1}$$

(Mariotti et al., 1981; Sigman et al., 1999). The isotope effect ε is defined as positive if δ$^{15}$N of remaining nitrate
increases when nitrate concentration decreases. The initial concentration and isotopic composition refer to that of the
nitrate in the water at the beginning of the growth season, which in our case is assumed to be retained in the Winter
Water layer (defined in Sect. 3.4).
Generally, the δ$^{15}$N of nitrate negatively correlates with its concentration: there is a visible correspondence between
low surface nitrate concentration (Fig. 6a) and high δ$^{15}$N of nitrate (Fig. 6b). δ$^{15}$N is low (4.5 to 5 ‰) in waters deeper
than 125 m, where nitrate concentration exceeds 30 μmol kg$^{-1}$, and reaches high values of up to 6.8 ‰ near the surface
in DP where nitrate concentrations are the lowest of these transects (23 μmol kg$^{-1}$). This negative correlation is
consistent with previous work, and reflects the isotopic fractionation during partial uptake of the available nitrate
(Altabet, 2006; Fripiat et al., 2019; Lourey et al., 2003; Sigman et al., 1999). While the nitrate reservoir is not strictly
isolated in summer, exchanges between surface mixed layer and subsurface waters are limited due to the density
gradient preventing wind-activated mixing (Lewis et al., 1986; Pollard et al., 1973). When approximating surface
waters above the MLD as a closed system, fractionation during nitrate uptake follows a Rayleigh-type isotopic
distillation (Lourey et al., 2003). In the Rayleigh distillation, the $\delta^{15}N$ is linearly correlated with the logarithm of
nitrate concentration (DiFiore et al., 2009; Mariotti et al., 1981). Given the high correlations at each station between
$\delta^{15}N$ and the logarithm of nitrate concentration (Pearson $r^2 > 0.9$ for all stations except LB07 with $r^2 = 0.633$; Fig.
A2), the approximation with Rayleigh distillation seems appropriate for the transects presented here.
The Rayleigh distillation model may not be applicable if changes to the isotopic composition are issued from processes
other than nitrate assimilation. Nitrogen recycling through heterotrophic activity may modify the isotopic repartition
between organic and dissolved inorganic nitrogen (Sigman and Fripiat, 2019). Recycling of N through nitrification
preferentially converts $^{14}N$ back to nitrate and may decrease the apparent $\varepsilon$ of N uptake (DiFiore et al., 2009). In the
summer, however, nitrate supplied by nitrification accounts for less than 10 % of nitrate uptake (DiFiore et al., 2010;
Flynn et al., 2021; Mdutyana et al., 2020). The nitrogen isotope effect $\varepsilon$ in KARP-20 profiles (Fig. 8) is close to
previously reported values for $\varepsilon$, so they are in line with low nitrate recycling and the approximate validity of Rayleigh
distillation. Light inhibition may underly this lack of nitrification in summer (Mdutyana et al., 2020).

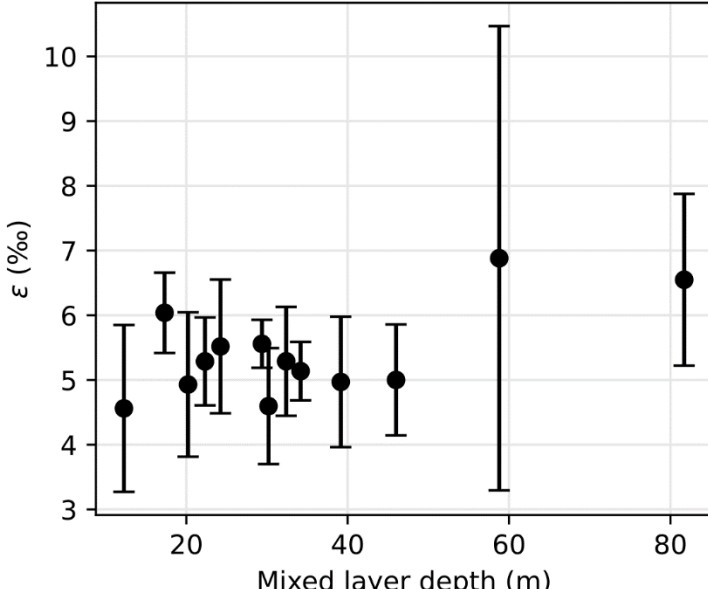


**Figure 8. Nitrogen isotope effect $\varepsilon$ as a function of the mixed layer depth. Error bars indicate the 95 % confidence interval**
**for $\varepsilon$ estimation.**
Nitrate assimilation has been reported to imprint a nitrogen isotope effect of about 5 ‰ in the SO (DiFiore et al., 2009,
2010; Fripiat et al., 2019; Sigman et al., 1999). Some studies have attributed greater isotope effect to continuous active
pumping of nitrate to offset nitrate loss by diffusion through cell membrane, when assimilatory nitrate reduction is
slowed by light-limited cellular activity (Needoba et al., 2004; Needoba and Harrison, 2004). In the SO, light limitation
was proposed as a cause of higher isotopic effects in waters with deeper mixed layers, with values exceeding 8 ‰
(DiFiore et al., 2010), but that finding must be reassessed given the recently recognized role for nitrate-nitrite N isotope
exchange in SO waters (Fripiat et al., 2019; Kemeny et al., 2016), which can lead to overestimation of $\varepsilon$ if the samples
have lost nitrite (e.g., in the case of sample preservation by acidification). Here, we quantify the isotopic effect of
nitrate uptake following the linear regression on logarithmic concentration scale method (DiFiore et al., 2009;
Appendix Fig. A2), station by station to evaluate spatial changes in fractionation effect during assimilation of nitrate
by phytoplankton. Consistent with Fripiat et al. (2019), we find nitrogen isotope effects $\varepsilon$ around 5 ‰, insensitive to
MLD for MLD ≤ 50 m (Fig. 8). It is possible that still deeper MLDs are associated with higher $\varepsilon$, but the weakness of
the regressions make this uncertain (Fig. 8). In any case, lower surface nitrate $\delta^{15}N$ and higher surface nitrate
concentrations around 61 °S in the LB transect are in line with environmental limitation of nitrate assimilation, that
we attribute to light limitation as MLD is larger at this location.
In summary, our data support that the relationship between nitrate depletion and $\delta^{15}N$ elevation, previously described
in other regions of the SO (DiFiore et al., 2010; Lourey et al., 2003; Sigman et al., 1999), holds true in the ice-covered
region of the Antarctic Peninsula. The consistency of the data with the Rayleigh model implies that nitrate resupply
did not occur late in the summer season, as this would have violated the Rayleigh model and largely removed the
surface nitrate $\delta^{15}N$ elevation.

### 4.2.3 Nitrogen isotope modelling

In this final section, we use a nitrogen isotope box-model (Yoshikawa et al., 2005) to reproduce the observed
differences between three basins of the studied region, which have different sea ice cover duration, and compare them
with KARP-20 nitrate patterns. Modelling the nitrogen cycle allows understanding of the seasonal evolution of nitrate
concentrations, whereas observations are usually made in a short time frame.
The model simulates a nitrogen cycle in a sea ice covered ocean, focusing on surface (0-20 m) and subsurface (20-
120 m) layers, from inputs of seawater temperature, mixing depth, insolation, and nitrate concentration and its $\delta^{15}N$
at the 120 m boundary condition (Fig. 9). The model has a daily time resolution with a simulation length of one year
(after 4 years spin-up), and reproduces an ideal annual cycle at equilibrium, *i.e.* repeating itself given the same forcings.
We run three simulations in oceanic locations representative of distinct basins, each with specific ice cover: the
seasonally ice-covered part of DP (60°S, 60°W), semi-enclosed eastern BS (62°S, 55°W), and the northwestern WS
(64°S, 50°W) where sea ice exits the Weddell Gyre.

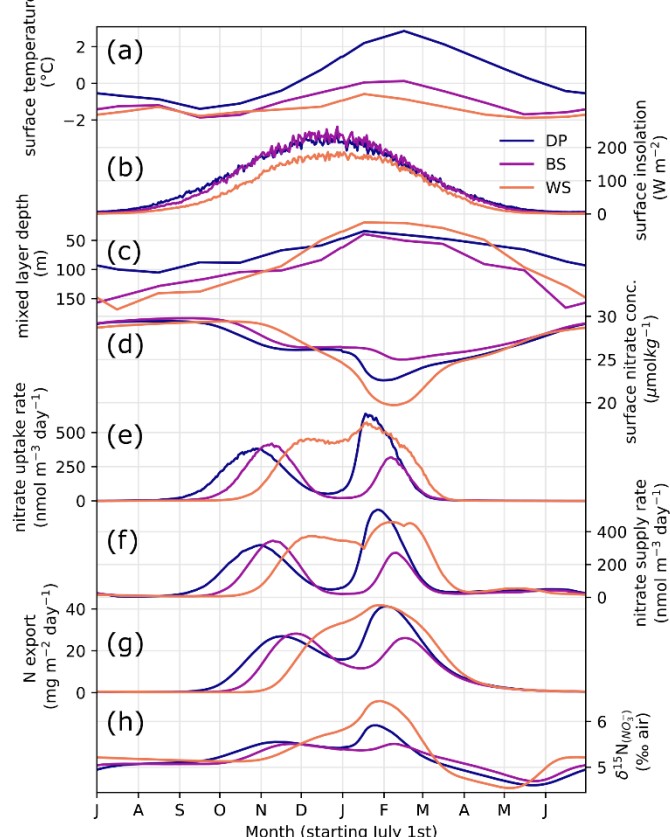


**Figure 9. Seasonal variability of model forcings and results: surface water temperature (a), insolation (b), MLD (c), surface nitrate concentration (d), nitrate uptake rate in surface model box (e), nitrate supply rate from subsurface to surface (f), sinking particulate N export (g), and $\delta^{15}N$ of nitrate in the surface model box (h) in three oceanic locations (DP, BS, and WS).**

We first give a brief description of environmental parameters used to constrain the model. Temperature variability is strongest in the DP location (60 °S, 60 °W), which averages 1.7 °C in summer, when BS reaches a maximum of around 0°C (Fig. 9a, Fig. 10a). The temperature at the WS location is buffered by the presence of sea ice, and remains below -0.5°C year-round. Insolation is roughly similar for the three sites, although slightly lower in the WS due to the higher latitude. MLD follows the typical seasonal variability (Behera et al., 2020), shoaling in the spring and summer and deepening during autumn and winter. It is implemented indirectly in the model, regulating the exchange rates between surface and subsurface layers, rather than modifying the layer thickness in the model. The amplitude of MLD variability is greater for BS and WS, where more sea ice is present (Fig. 9c, Fig. 10d). The MLD in BS remains deeper even in summer, averaging 59 m. BS location has an intermediate temperature and yearly sea ice (Fig. 10a, 10b), but also the greatest sea ice standard deviation (Fig. 10c), used to approximate the seasonal variability of sea ice cover here. Despite greater change in sea ice cover, the summer MLD is still greater in BS location, which goes against the hypothesis of a sea ice meltwater induced summer stratification (Taylor et al., 2013), and confirms that sea ice seasonal variability is not the main control on summer MLD in this region (as discussed in Sect. 4.1).

At the WS location, growth season is shorter due to its onset delayed by the late melting of sea ice, resulting in a single yearly maximum rate of nitrate uptake. Nevertheless, WS has a greater nitrate depletion (Fig. 10j) in the surface layer compared to the two other locations, driven by higher nutrient uptake in the strongly stratified surface ocean (Fig. 10f). The BS location with the deepest mixed layer is characterized by a lower nitrate uptake especially in the later part of summer (Fig. 9e), resulting in a lower depletion and higher concentration relative to the two other locations. $\delta^{15}N$ of nitrate (Fig. 10i) scales with the net nitrate uptake (Fig. 10f-h), with maximum $^{15}N$-enrichment at WS location, and minimum at BS location. This matches observations of higher nitrate concentrations in summer north of the Antarctic Peninsula, and around 61 °S in the LB transect. The model also predicts that N export due to sinking particulate organic matter follows the same pattern (Fig. 10k).

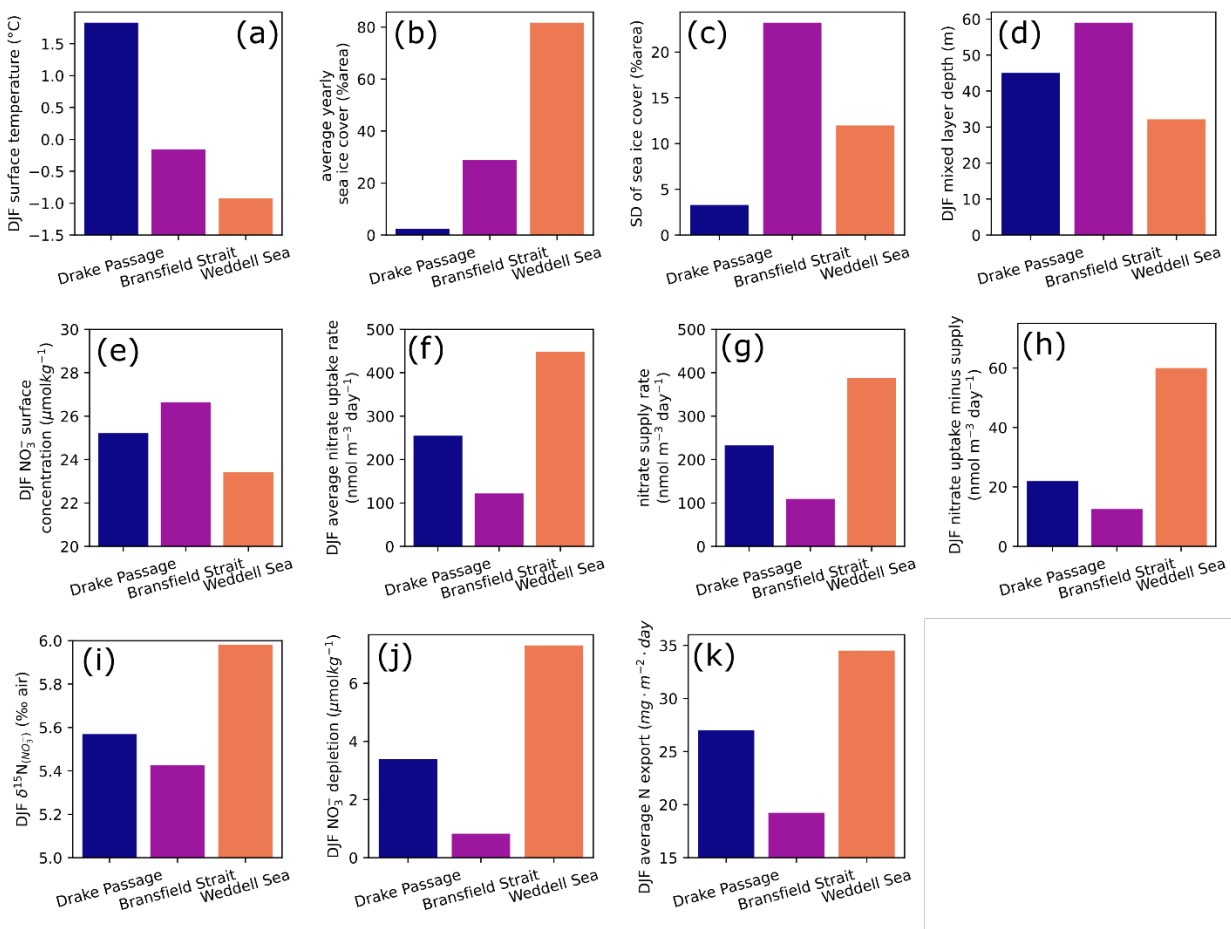

**Figure 10. Summer (December, January, and February) averages of model forcings and results for (a) surface water temperature, (d) mixed layer depth, (e) surface nitrate concentration, (f) nitrate uptake rate by primary producers, (g) nitrate supply rate from subsurface to surface model layer, (h) difference between uptake and supply rates, (i) $\delta^{15}$N of nitrate in surface water, (j) nitrate depletion in the surface water, (k) total nitrogen export in the sinking flux in three oceanic locations. Yearly sea ice parameters are given for (b) average and (c) standard deviation, used here to quantify seasonal variability. Upper row plots (a-d) are environmental forcings, with temperature (a) and mixing depth (d) forcing the model. Lower rows (e-k) describe nitrogen cycle model results.**

Both in the model results and in the latitudinal transect (Sect. 4.2), it appears that the eastern BS has the lowest productivity due to its deeper mixed layer, despite intermediate temperature and sea ice seasonality. The higher surface nitrate concentration of the model for BS matches the 61–62 °S section with high surface nitrate concentration (Fig. 6), although absolute values are slightly lower in the model. Timing-wise, the model tends to predict that surface nitrate concentration at BS is most different from other locations during the later part of summer (around February, Fig. 9), but December observations of nitrate concentration are already marked by strong latitudinal differences (Fig. 6a). The area of low nitrate uptake east of the SSI matches the low chlorophyll area in the climatology of surface ocean color (La et al., 2019). The location of BS point chosen for the model may not be exactly in this low chlorophyll area but closer to the Peninsula Front. However, model results should not differ because the two are characterized by a relatively deep summer mixing and similar ice cover. In addition, the eastern BS is notably characterized by a deepening of isopycnals (Frey et al., 2022; Huneke et al., 2016) and deeper chlorophyll maximum (Russo et al., 2018), attesting that the MLD increases eastward, and primary producers are likely mixed to deeper waters.

In the SO, despite low temperature, a significant portion of particulate organic matter is remineralized in the water column, and nitrogen loss in surface water by sinking particles is compensated by the upwelling of nutrients, although seasonality of these fluxes is not in phase (Mdutyana et al., 2020). Here, we focused on the summer season, therefore

the net export of nitrogen that we report does not represent an annual flux. However, the model indicates that
productivity decreases with deeper mixing, in line with our interpretation of transect data, and this translates to reduced
flux of sinking particles in the summer.
Overall, the model confirms the observations that sea ice seasonality is not the principal control on productivity and
surface nitrate drawdown, which are rather tied to summer MLD. The $\delta^{15}N$ of nitrate increases with surface nitrate
drawdown, and both vary jointly with net productivity and particulate export.
**5. Conclusions**
We investigated nitrate dynamics near the Antarctic Peninsula in the Southern Ocean, to better understand if sea ice
impacts primary productivity by analyzing nitrate drawdown. We compiled measurements from different databases
and new transects for a total of 394 nitrate profiles. In this region seasonally covered by sea ice, nitrate is not limiting
productivity, and remains in concentrations above 20 μmol kg$^{-1}$ at any time of the year. We evaluated the influence of
sea ice on nitrate depletion in the surface water, testing the hypothesis that sea ice melting reduces surface salinity and
enhances the stratification propitious to primary producers (Taylor et al., 2013). We used nitrate depletion, defined as
surface concentration minus concentration in the Winter Water layer, as an indicator of nutrient uptake and net
seasonal productivity. Results do not clearly point to a change in nitrate depletion in regions where sea ice duration
differs. Significant nitrate depletion is mostly observed after melting, owing to favorable conditions for blooming after
ice melt. However, sea ice melting is not necessarily followed by nitrate depletion. Seasonal amplitude of sea ice only
marginally affects nitrate depletion and stratification, opposing the hypothesis that sea ice meltwater controls
stratification at the regional scale. Diverse oceanographic controls in the water masses properties, density gradients
and mixing may mask the impact of sea ice on productivity and nitrate depletion when comparing a variety of locations.
Analysis of nitrate along new north-south transects in the DP and Powell basin east of the BS reveals a channel of
deeply mixed waters with high surface nitrate concentration. We interpret this channel to be a remnant of actively
homogenized water with tidal mixing along the SSI upstream the BS (Zhou et al., 2020). $\delta^{15}N$ of nitrate in these
transects confirm the previously established relationship between nitrate drawdown and $^{15}N$-enrichment in the
remaining nitrate. The strong correlation between logarithm of nitrate concentration and $\delta^{15}N$ indicate that the system
is well approximated by a Rayleigh distillation, with limited resupply of nitrate during phytoplankton growth. Nitrogen
isotope modeling further supports that deeper mixing in the BS induces light limitation, with lower nitrate uptake and
$\delta^{15}N$ compared to both the DP and the WS. Weaker sinking particle flux in the eastern BS equates to weaker organic
matter export in summer.
Strong stratification in the summer surface ocean makes nitrate depletion a persistent summertime feature, useful for
integrating biological uptake over a growth season. However, variations in MLD make it difficult to quantify nitrate
uptake and productivity based on surface concentration changes alone, requiring vertical profiles of nitrate
concentrations. Nonetheless, the consistency of the nitrate data with the Rayleigh model imply that there were was
not a strong resupply of nitrate late in the summer season, such that surface nitrate concentration changes are a
plausible indicator of spring-summer nitrate assimilation..
Increasing temperatures with global climate change will likely reduce the extent of sea ice in the future. Summer
stratification could shift from a salinity-based density gradient to a temperature-based density gradient. This will
probably impact phytoplanktonic activity and ecosystems in a warmer surface layer with a longer ice-free season. A
supposed decrease in freshwater supply in the surface ocean would tend to increase MLD inducing light limitation on
phytoplanktonic productivity, but this would be counterbalanced by increased productivity in warmer waters and
longer growth season. Determining which of these two effects prevails is crucial to understand future changes in the
phytoplankton productivity of the high latitude Southern Ocean.

**Appendix A: Station-specific profiles and isotope effect**

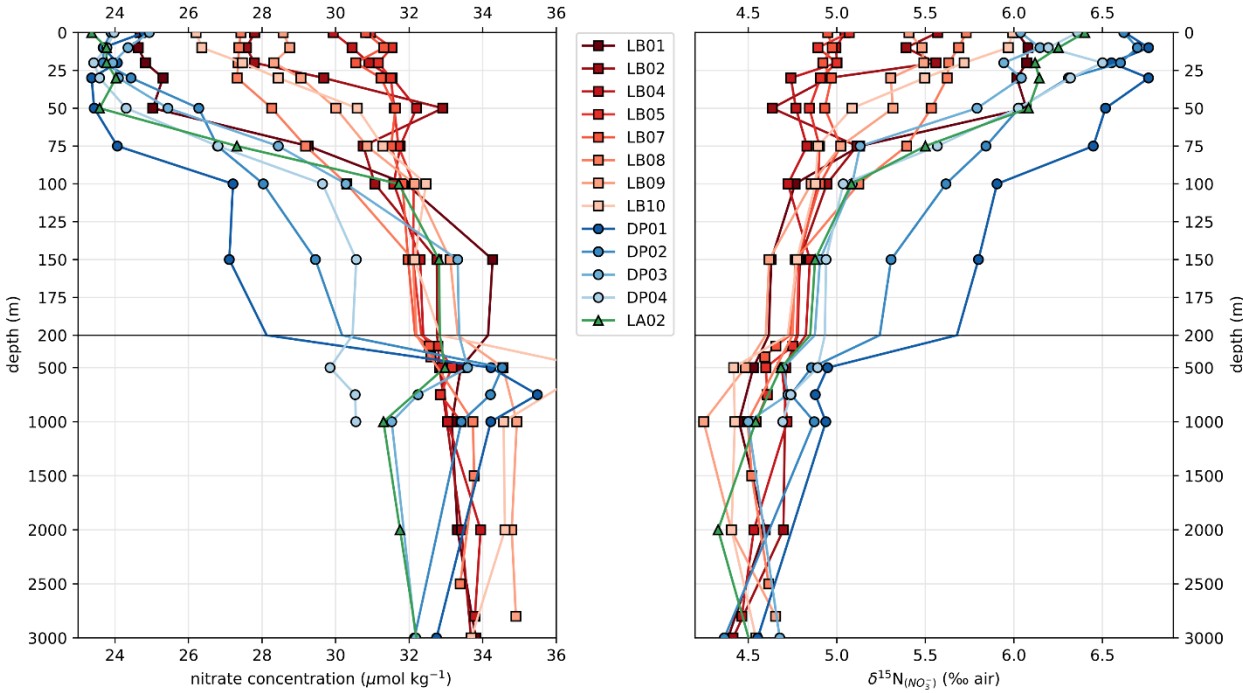


**Figure A1. Station-specific profiles of (a) nitrate concentration and (b) $\delta^{15}$N of nitrate.**

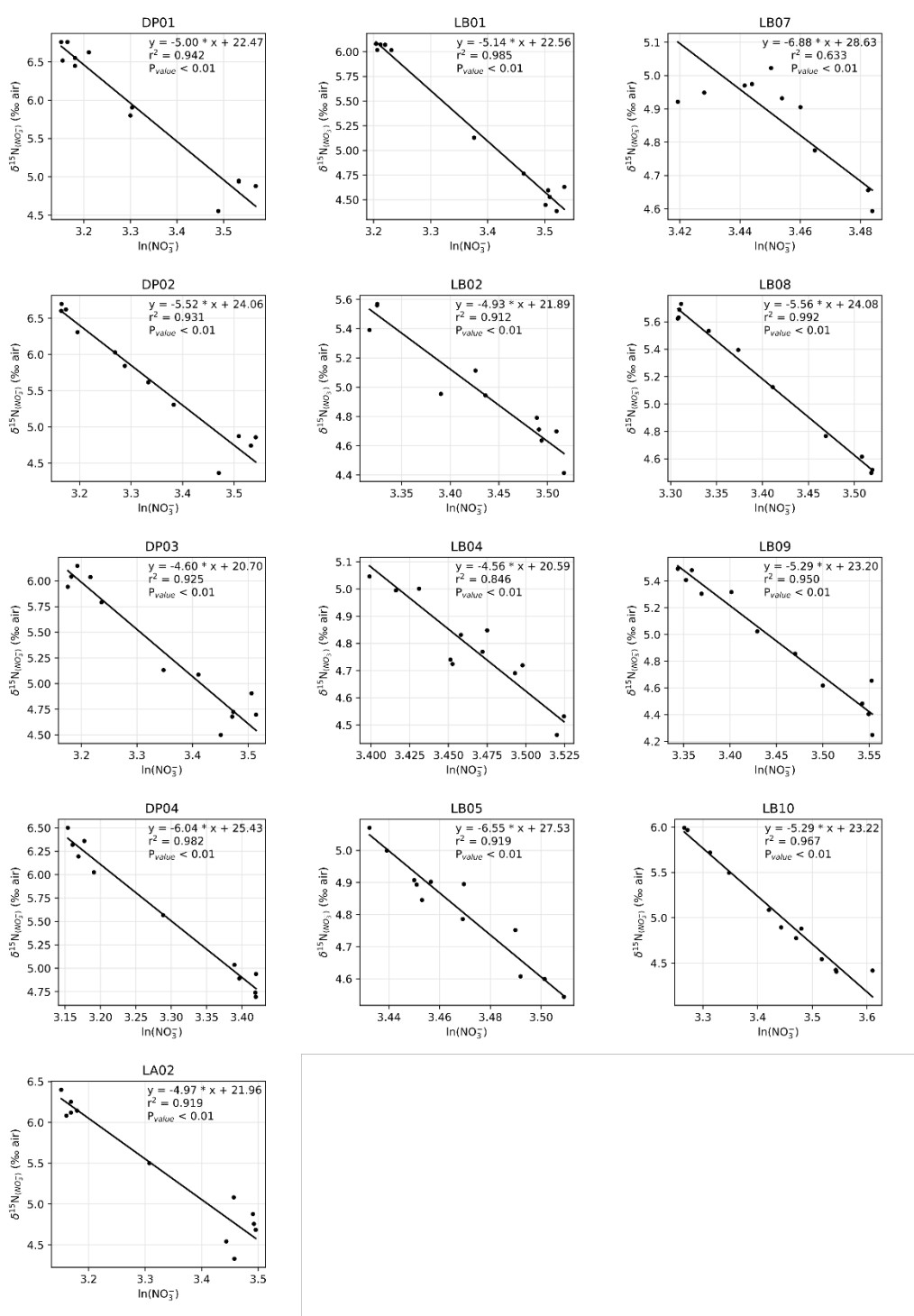



**Figure A2.** $\delta^{15}N$ of nitrate as a function of nitrate concentration (logarithmic scale) for each KARP-20 profile, used for estimation of nitrate isotope effect $\varepsilon$ (opposite of the slope value, Eq. (1)).

**Code availability**

Python code for data processing and figure creation is available on a Zenodo repository (doi:10.5281/zenodo.14221104) for transparency and reproducibility purposes. Potential users should know that it was developed for personal use and has not been cleaned up before distribution. Nitrogen cycle model inquiries should be sent to C. Yoshikawa (yoshikawac@jamstec.go.jp).

**Data availability**

KARP-20 data inquiries should be addressed to B.-K. Khim (bkkhim@pusan.ac.kr), and data will be distributed after evaluation of the request. GLODAP bottle data is available for download at https://glodap.info/ [last access: November 2024]. SOCCOM float data is available for download at https://soccom.princeton.edu/float-data-single-float-profiles [last access: November 2024]. Sea ice concentration data is available for download at https://osi-saf.eumetsat.int/ https://glodap.info/ [last access: November 2024].

**Sample availability**

Samples from this study are not available.

**Author contribution**

Concept of the study, formal analysis and figure creation were conducted by APMS, as part of project co-led by FJJE and NO. The nitrogen model was developed by CY. Water samples were collected by YJ and BKK. Nitrogen data was obtained and curated by YJ, BKK, YR and DMS at Princeton university. The original manuscript prepared by APMS, YI, and CY, with revisions from BKK, NOO, YR, FJJE and NO.

**Competing interests**

The authors declare that they have no conflict of interest.

**Acknowledgements**

Float data were collected and made freely available by the Southern Ocean Carbon and Climate Observations and Modeling (SOCCOM) Project funded by the National Science Foundation, Division of Polar Programs (NSF PLR-1425989, with extension NSF OPP-1936222), and by the Global Ocean Biogeochemistry Array (GO-BGC) Project funded by the National Science Foundation, Division of Ocean Sciences (NSF OCE-1946578), supplemented by NASA, and by the International Argo Program and the NOAA programs that contribute to it. The Argo Program is part of the Global Ocean Observing System (doi:10.17882/42182, https://www.ocean-ops.org/board?t=argo). We thank the GLODAP project members who maintain and update the quality-controlled discrete sample database. The data from the EUMETSAT Satellite Application Facility on Ocean and Sea Ice used in this study are accessible through the SAF's homepage (https://osi-saf.eumetsat.int/). We are grateful to Thomas Lavergne for guidance and updates of the sea ice product. We would like to thank the members of the KARP-20 research expedition for collecting samples and making this work possible.

**Financial Support**

Collection and analysis of samples presented in this study were supported by Korea Institute of Marine Science & Technology funded by the Ministry of Oceans and Fisheries (RS-2023-00256330: to BKK).

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
