# Peer review of "Sea ice and mixed layer depth influence on nitrate depletion and associated isotopic effects in the Drake Passage – Weddell Sea region, Southern Ocean"

_EGUsphere, 2024_

## Author Response (AR1)

**Author Comments on egusphere-2024-3687**

https://egusphere.copernicus.org/preprints/2024/egusphere-2024-3687/

**Reviewer comments** **in black**

**Author answers** **in red**

Prepared by A. Servettaz, with contributions from co-authors. Last update February 20.

**Response to Referee Comment #1**

**General comments**

Motivation/coherence

While I'm not an expert on isotope chemistry, I don't see any obvious flaw in the
methodologies of the paper. It is very descriptive, but that is okay as long as the overall
motivation is clear. However, the motivation isn't always clear, and it becomes somewhat
less clear as more parts of the analyses are being discussed. My concern is that the individual
pieces of the analysis aren't well connected. Or rather: that it's not made very clear how they
are connected, and why they're needed to answer the main questions.

We changed the description of objectives to questions that we have actually treated, and
rearranged the order of introduction for a better logical progression:

We assume that nutrient depletion can be used to describe net primary productivity
(introduction, lines 46-63, unchanged).

We focus on nitrate and introduce N-cycle (moved up from 86-95). We complemented this
paragraph with "*We aim to evaluate the environmental factors that control nitrate depletion,*
*and by extent the net productivity*"

We describe the known limitations on productivity (lines 64-85, moved down accordingly).
The paragraph is now introduced with: "In the SO, several environmental factors can limit
phytoplankton growth and primary productivity." We completed the paragraph on iron
limitation (see the response to the corresponding comment below)

We draw the reader's attention to the fact that Bransfield Strait is different from most of the
SO because it is not iron limited, and therefore subject to other limitations notably related to
sea ice. (further justified in the reworked Section 2. Oceanographic Setting, see the response
to the corresponding comment below).

Is the isotope data mainly confirming that the approach used for the nitrogen deficit was
valid (L436-438)? Or is it mainly helping to understand what's happening in the deep mixed
layers in the Leg B transect, hinting at light limitation? Or both? I'm also not sure I
understand what the model adds to the overall interpretation of the data. So I recommend
making the connections and motivations clearer, leading the reader through the story that the
data and analyses tell.

The interest of studying isotopes was poorly introduced, first described in the corresponding
section. We add a brief introduction to why we are looking at them and what are the
objectives in this work:

"*Isotopes can be used to track environmental processes. They are used, for example, in*
*paleoenvironment studies to infer changes that occurred in the past (e.g. using N isotopes:*
*Studer et al., 2015). Knowing how nitrogen isotopes relate to nitrate concentration in the*

*modern ocean opens the use of nitrogen isotopes as a tracer of past nitrate changes.*
*Therefore, we study the link between nitrogen isotopes and nitrate concentration using both*
*observational data and isotope-enabled simulations. To this end, we describe in further detail*
*a previously unpublished transect of the concentration and nitrogen isotopic composition of*
*nitrate, interpreted with the help of model simulations in three oceanic locations.*"

Moreover, we added a short transition at the beginning of Sect. 4.2. to transition to the
isotopic aspects: *"While nitrate concentration does not appear clearly linked to sea ice,*
*investigating the isotopic signature in nitrate may reveal further constraints on nitrate*
*uptake. Additionally, how the nitrogen isotopes relate to nitrate concentration and other*
*environmental processes is necessary information to be able to use isotopes as a tracer of*
*these processes."*

Changes to the abstract have been made accordingly.

Additionally, if some of the analyses are merely supporting main conclusions coming from
one aspect of the work, the sections pertaining to the supporting analyses could maybe be
shortened/more focussed.

As we rewrote motivation and transitions, we hope that the necessity of each section appears
more clearly. Some sections in the discussion have been shortened.

**Specific comments**

Why this region?

The general question of what influence sea ice exerts over nitrate uptake is a worthwhile one
to try and answer, but if one was interested in a universal answer, the analysis should
encompass the whole sea ice zone in the Southern Ocean, not just this area around the
Antarctic Peninsula. So why the focus on this area, what is special about it? If anything, it
seems like an oceanographically complex region, so whatever the answer regarding the
influence of sea ice that is found here, it's likely not going to be universally applicable.
That's fine, but it would be good to motivate more clearly the interest in *this question* in *this*
*region*.

As the reviewer pointed out, the complex oceanographic region makes the conclusions found
in this article difficult to apply for other regions. During preliminary work for this article, we
tested wider regions but the relationship between sea ice and nitrate depletion became less
clear as we expanded the area. One solution would have been to separate different regions to
explain the entire sea ice zone. However, we chose our study area to focus on the region
around the Antarctic Peninsula, because it is most relevant to the data described in Sect. 4.2,
and for the reasons added below.

The following justifications for focusing on BS will be added in the introduction of Sect. 2
Oceanographic setting:

*"While some studies have described the relationship between sea ice and phytoplankton*
*development in other regions of the SO (e.g. von Berg et al., 2020; Briggs et al., 2018; Taylor*
*et al., 2013), we focus here on the region around the northern tip of the Antarctic Peninsula,*
*from DP to WS, with a particular focus on the BS. West Antarctic Peninsula and BS are*
*coastal regions with non-depleted surface iron (Jiang et al., 2019), meaning that*
*phytoplankton growth is more susceptible to be limited by light, as modulated by shading and*
*stratification processes related to ice cover and its melt. Moreover, the West Antarctic*
*Peninsula is the region with the fastest decreasing trend of ice concentration (Jones et al.,*
*2016), raising the question of how productivity will change with decreasing ice cover."*

What about iron limitation?

I agree that nitrogen data can be used in iron-limited regions to learn about productivity,
there's no problem there. However, the first sentence of the conclusions states that the goal of
this study was to better understand environmental controls on primary productivity - but iron
limitation was not investigated at all. In the introduction the authors state that the Bransfield
Strait isn't iron-limited, but the area investigated is much larger than the Bransfield Strait,
and iron limitation is very likely to play a role in this larger area. So any investigation of the
"environmental drivers" that doesn't consider iron limitation is missing a big potential piece
of the puzzle. Indeed, the fact that the sea ice seasonal range does have a measurable
influence on nitrate drawdown but the MLD is not the driver (Figure 5) could be interpreted
as an indication that another aspect of sea ice melt, for example release of dissolved iron, may
be responsible for the observed relationship. So I recommend that the authors reconsider, and
discuss in more depth, the potential influence of iron limitation on productivity in the region.

Mentioning "environmental drivers" in the conclusions was too vague, compared to what we
could achieve with the present data. We changed it to: "We investigated nitrate dynamics near
the Antarctic Peninsula in the Southern Ocean, to better understand if sea ice impacts primary
productivity by analyzing nitrate drawdown."

We also clarified other instances of 'environmental parameters' by using specific terms such
as nitrate drawdown or net productivity.

While an actual study of environmental drivers would be of interest to a broader public, the
data presented here cannot sufficiently answer these questions. The work we conducted here
was to clarify how nitrate depletion and nitrogen isotopes relate to productivity in regions that
are not N-limited. We introduced the limiting factors on biological productivity, of which
iron is the most areas of the Southern Ocean, but also point out the particularity that light is a
co-limiting factor, especially around the Antarctic Peninsula, thus we evaluate the potential
sea-ice driven light limitations.

In particular, iron data are lack in published databases to establish any significant link to sea
ice. We conducted preliminary testing with diverse macro- and micro-nutrients before
choosing to focus on nitrate for which high quality data is abundant. Only 70 dissolved iron
concentration data points in the study region are reported in the GEOTRACES database,
which did not correlate significantly to any of the sea ice indicators tested.

However, we completed our introduction section on iron limitations by referencing other
works for Drake Passage and Weddell Sea dissolved iron concentrations:

*"In the Drake Passage and offshore west peninsula shelf break, the dissolved iron content is*
*extremely low even at some locations over the continental shelf ($< 0.1$ nmol $kg^{-1}$), and thus*
*limits productivity (Annett et al., 2017). The Antarctic Peninsula side of the Weddell Sea*
*receives iron from melting icebergs, which contributes to dissolved iron concentration*
*slightly higher than the central Weddell Sea or the Drake Passage, especially over the*
*continental shelf where concentrations exceed 0.2 nmol $kg^{-1}$ (Klunder et al., 2014)."*

And concluded with:

"Light availability, controlled by ice shading and vertical mixing, limits phytoplankton
growth in these areas BS (Gonçalves-Araujo et al., 2015), which is especially true for winter
when light intensity is low (Hatta et al., 2013). This makes it an area of particular interest for
studying the relationship between sea ice cover and primary productivity."

In addition, we mentioned the potential contribution of melting sea ice to lifting iron
limitation:

(added in Sect 4.1.2) *"Melting of sea ice can also favor phytoplankton bloom by releasing*
*iron that concentrates in sea ice (Boyd and Ellwood, 2010; Lannuzel et al., 2016), in*
*particular for regions where iron is limiting such as the DP."*

(added in 4.1.3) *"Possible iron supply from ice melt may also increase productivity without*
*changing MLD (Lannuzel et al., 2016), and may partly explain why nitrate depletion is better*
*correlated to $SD_{SIC}$ than MLD."*

but complex interactions between ice melt iron release, uptake and remineralization by
biological processes, or other physical removal of iron were not the topic of this article, and
we do not have supporting data to discuss it.

Some comments in order of appearance:

L 19: Since nitrate is not limiting in the region, this should probably refer to the re-supply of
iron? See also comment about iron from sea ice melt above.

We replaced nitrate with the less specific term "nutrient".

L 52: Argo data are now allowing us to look under the sea ice, so satellites are not the only
means to remotely measure the ocean under ice.

I used remotely in the sense of remote-sensing. Argo are doing automated *in situ* sensing. I
clarified the sentence in that regard: "ability to evaluate the productivity of seasonally ice-
covered regions by remote-sensing methods"

L 186: Refer to figure 3 here? This would help the reader understand that these analyses are
used for the maps in this figure

We added a reference to figure 3.

L 186-194: Consider citing some more recent studies that have used this approach with the
WW (e.g. Moreau et al.)

Added a citation to Moreau et al. (2020) along Goeyens et al. (1995) at L188. We would like
to point out that our method is most similar to Flynn et al. (2021), which is cited L190.

L 242-245: Aren't the two sentences basically saying the same thing? Maybe this could be
streamlined.

We merged the sentences together and simplified:

*"We did not backtrack either the water parcel or the sea ice, assuming that the seasonality of*
*ice at the point of measurement (fixed location) resembles that of the water parcel in which*
*nitrate was measured (moving parcel)."*

L 294-295: I can't see how this summary is derived from the Figure.

The sentence was corrected to "nitrate depletion is generally greatest about 70 days after ice
has melted" and the reference was corrected to figure 4b. In addition, we added the 25-day
rolling mean for better readability. The value was adjusted to 70 days to match the maximum
in average nitrate depletion read on rolling mean.

Figure 4b: How is ice melt defined here? When sea ice concentration is 15% or less? Please
specify in the Figure caption.

We added the definition: *"defined as the number of days spent with ice concentration below*
*15% (or days left until this threshold is crossed for negative values)"*

L 316-317: Is it worth also testing other metrics for seasonal changes in sea ice cover (other
than the standard deviation)? For example, the amplitude (between max and min sea ice
cover) could be a metric to try.

We tested other definitions, notably maximum minus minimum, and average sea ice. We
considered amplitude to be unreliable because it uses the measurement for two dates rather
than the full annual cycle, and is thus more susceptible to analytical errors or may even
especially highlight outliers that have not been eliminated during sea ice concentration data
processing. Moreover, the correlation with nitrate concentration or nitrate depletion was
lower than for standard deviation. For average sea ice concentration, it was ruled out because
it does not necessarily imply high meltwater input, especially for Weddell Sea point where
sea ice does not completely melt in summer.

L 327: Please add "in this region" to the end of the last sentence.

Added the phrase as suggested.

L 328, 331: When saying SIC here, do you mean variability (std) of SIC? That's what the figure shows that is being referred to.

Yes, we added *"SIC standard deviation (SD)"*. We homogenized standard deviation notations to SD in figure as well, as per Copernicus recommendations.

L 354 (Eq 1): What is taken as the initial here?

We added the following below (Eq. 1):

*"The initial concentration and isotopic composition refer to that of the water at the beginning of the growth season, which in our case is assumed to be retained in the Winter Water layer (defined in Sect. 3.4)."*

L 364-366: Can these two goals really be mixed? If there are deviations from the expectation, how can you tell whether that has to do with environmental forcings versus a violation of the relationship? I have trouble imagining what it would mean/look like if "the relationship between nitrate concentration and isotopes" were not respected. So maybe it's my ignorance but if that's the case, then a few explanatory words might help.

This comment and the following ones highlighted that several subjects were treated indistinctly in this section. For better readability, we decided to separate the KARP-20 analysis section into two:

4.2.1 KARP-20 nitrate concentration and $\delta^{15}$N transects, where we lightly describe the concentrations and oceanographic properties

4.2.2 $^{15}$N enrichment during Rayleigh distillation, where we explain the isotopic signal acquisition, and why we can use a Rayleigh distillation model.

Former discussion about light limitation and their effect on isotopic fractionation, mostly based on DiFiore et al. (2010), have been shortened as most recent study by Fripiat et al. (2019) as highlighted by Reviewer#2 contradicts our previous interpretation (see the response to RC2 for further details).

To answer the comment more specifically: the relationship between concentration and isotopic composition is approximated by a Rayleigh model, where two pools of nitrogen (Biology and Ocean) can exchange in a relatively closed system. In the Southern Ocean, this relies on stratification of the water column during growth season, and absence of nitrogen fixation or denitrification. If these conditions are not respected, i.e. a third pool comes into play (deep ocean in case of non-stratified column, or atmosphere in case of gas exchanges), the Rayleigh distillation model is not sufficient and models with more complexity are required to explain the isotopic exchanges. Given the high correlations in the Appendix Fig. A2, the Rayleigh approximation is deemed appropriate (the points are aligned).

We rephrased the section goals to clearly state that we want to verify that we can approximate our system with a Rayleigh distillation:

*"While nitrate concentration does not appear clearly linked to sea ice, investigating the isotopic signature in nitrate may reveal further constraints on nitrate uptake. Additionally, how the nitrogen isotopes relate to nitrate concentration and other environmental processes is necessary information to be able to use isotopes as a tracer of these processes. In this section, we focus on the DP and BS transects (location given in Fig. 1), with measurements of nitrate concentration and $\delta^{15}N$. We verify that the relationship between nitrate concentration and isotopes follows a Rayleigh distillation in this region (Sect. 4.2.2). We then validate our interpretations against an isotope-enabled nitrogen cycle model forced by environmental parameters taken from three characteristic sub-regions (Sect. 4.2.3)."*

We completed the Eq. 1 with better introduction to Rayleigh distillation, and moved it to the corresponding Section 4.2.2:

*"A system where a component is progressively removed without coincident resupply can be described by the Rayleigh distillation model. It can be applied to describe isotopic fractionation during nitrate uptake in the condition that there is net removal from the water through one process (nitrate assimilation), and the removed nitrate has an isotopic composition that differs from that in the water. In this case, the isotope effect can be quantified by the following approximation using measurable parameters:"*

Possible sources of discrepancies with Rayleigh model issued from nitrogen recycling are discussed in the following paragraph, which has been moved up from L427-435, and introduced with:

*"The Rayleigh distillation model may not be applicable if changes to the isotopic composition are issued from processes other than nitrate assimilation."*

We also concluded on its relevance in our study:(added in conclusion of 4.2.2) *"Given the high correlations at each station between $\delta^{15}N$ and the logarithm of nitrate concentration (Pearson $r^2 > 0.9$ for all stations except LB07 with $r^2 = 0.633$; Fig. A2), the approximation with Rayleigh distillation seems appropriate for the transects presented here."*

L 406-407: For someone new to this type of analysis, a bit more detail about how to think of this analysis/interpret it would be useful here. Why is this analysis (Figure A2) done station by station, what's the reasoning for that? If there's a disconnect assumed between MLD and the deeper waters, it's not obvious to me that the deep data at a given station should be used? Does the noise around the slope hold any information that is worth interpreting, or is it just the slope (e.g. LB07, which is also one of the 2 stations with the high epsilon)?

Two pieces of information can be derived from Appendix Figure A2. First is that the points align, which is in line with a Rayleigh distillation, as discussed in the answer to previous comment. Second, the value of the regression slope can be a marker of changes in the nitrogen cycle. This sentence (L406-407) corresponds to the second interpretation that has been moved to the corresponding paragraph, and completed with an explanation of why the analysis is done for each station:

*"Nitrate assimilation has been reported to imprint a nitrogen isotope effect of about 5 ‰ in the SO (DiFiore et al., 2009, 2010; Sigman et al., 1999). Some studies have attributed greater isotope effect to continuous active pumping of nitrate to offset nitrate loss by diffusion through cell membrane, when assimilatory nitrate reduction is slowed by light-limited cellular activity (Needoba et al., 2004; Needoba and Harrison, 2004). In the SO, light limitation was proposed as a cause of higher isotopic effects in waters with deeper mixed layers, with values exceeding 8 ‰ (DiFiore et al., 2010), but that finding must be reassessed given the recently recognized role for nitrate-nitrite N isotope exchange in SO waters (Fripiat et al., 2019; Kemeny et al., 2016), which can lead to overestimation of ε if the samples have lost nitrite (e.g., in the case of sample preservation by acidification). Here, we quantify the isotopic effect of nitrate uptake following the linear regression on logarithmic concentration scale method (DiFiore et al., 2009; Appendix Fig. A2), station by station to evaluate spatial changes in fractionation effect during assimilation of nitrate by phytoplankton. Consistent with Fripiat et al. (2019), we find nitrogen isotope effects ε around 5 ‰, insensitive to MLD for MLD ≤ 50 m (Fig. 8). It is possible that still deeper MLDs are associated with higher ε, but the weakness of the regressions make this uncertain (Fig. 8). In any case, lower surface nitrate $\delta^{15}N$ and higher surface nitrate concentrations around 61 °S in the LB transect are in line with environmental limitation of nitrate assimilation, that we attribute to light limitation as MLD is larger at this location."*

L 412-426: Could this paragraph be tightened to make the main finding (light limitation in the profiles with deep MLD) clearer and relate it back to the motivation stated on lines 394-395? Does this result then exclude the other explanation for the high nitrate concentration mentioned on lines 392-394? And could there be any other explanations for the high observed epsilon (other than light limitation)?

Comments from Reviewer#2 incited us to revise the interpretation of epsilon. This paragraph has been changed to the version given as a response to the previous comment. The full justification is given in the response to RC2.

L 440-446: Can the locations that were modelled please be shown in Fig 1 or Fig 2? Also, how/why were these specific locations chosen, what was the motivation?

The model remains relatively simple, and having a multitude of points would not be more informative. We wanted to compare among the three main basins in this region, so we took one point each in Drake Passage, Bransfield Strait and Weddell Sea. Locations of model points were added to Fig. 1.

L 503-504: What the model results show for the biological pump will depend highly on the parameterization. So I suggest to either cut that bit or lay out more explicitly what the parameterization for the biological pump was, and how well it's expected to work in this region.

Describing the biological pump would add length to an already substantial article. As biological pump is not discussed in detail at any point of this manuscript, we preferred removing its mention, and keep the focus on primary productivity and nitrate drawdown.

L 517-518: Is this one of the main results of this study? If so, can this be explored a bit more? How does this compare to what we already knew about the region?

This is the interpretation that we give to the low correlation between sea ice seasonality and nitrate depletion. However, it is not proven by the data presented in this study, so we changed the sentence to conditional. The corresponding discussion (L339-346) was completed with additional references:

*"Regional differences in productivity response to MLD were also found West of the AP (Vernet et al., 2008), and Chlorophyll a concentrations in the BS do not appear to correlate with MLD either (Romanova et al., 2021)."*

L 529-535: The nitrate deficit approach is hardly qualitative, it's quantitative. This whole paragraph could use more focus/tightening.

We removed unnecessary discussions about MLD in the second half of this paragraph, and clarified our point regarding the cases in which nitrate can provide quantitative/qualitative net productivity estimations:

*"Strong stratification in the summer surface ocean makes nitrate depletion a seasonal-lasting feature, useful for integrating biological uptake over a growth season. However, variations in MLD make it difficult to quantify nitrate uptake and productivity based on surface concentration changes alone, requiring vertical profiles of nitrate concentrations. Nonetheless, single surface nitrate concentration drawdown, or $\delta^{15}N$ that reflect concentration changes, may provide qualitative productivity estimations."*

**Technical corrections**

L 203-204: Consider rewording the sentence starting with "Therefore"

Changed to *"The amplitude of the dilution effect is too small to impact the conclusions of this article."*

L 216: Replace "international" with "interannual"?

corrected

L 263: Should this refer to Figure 3 instead of Figure 2?

Figure 2 also showed sea ice duration, but Figure 3 is more appropriate. The reference was changed to Fig. 3.

L 285: Consider replacing the word "non-segregated"

Changed to "well-mixed"

Fig 6: Consider switching the colour bar for the top panels so it matches the bottom? Darker
colours are also usually associated with higher concentrations, not lower.

The palette used here (viridis) typically has its highest value in yellow. For nitrate
concentration, see for example the use of the same palette in the Fig, 2 in Moreau et al.,
(2020). Regarding the consistency, it has been decided between the authors that "highest
value is in yellow" should be the consistent point between the panels, thus highlighting the
negative correlation between nitrate concentrations and its $\delta^{15}N$. We do not make any change
at this point, but are open if deemed necessary despite our argument.

L 491-492: Not sure what is meant by "divergence of nitrate concentration" and also what the
word "differs" refers to

This sentence was rephrased to:

"Timing-wise, the model tends to predict that surface nitrate concentration at BS is most
different from other locations during the later part of summer (around February, Fig. 9), but
December observations of nitrate concentration are already marked by strong latitudinal
differences (Fig. 6a)."

L 498: Phytoplankton can also grow at depth, and "advected" is probably a more appropriate
term here than "dragged"

Corrected as suggested.

**Response to Referee Comment #2**

This study examines the influence of sea ice and mixed layer depth on productivity in the Drake Passage and Weddell Sea regions—a significant and ongoing area of debate. While the manuscript is well-written and most analyses are sound, I find that the connections and added value of each section are sometimes unclear and redundant, which could be improved.

As suggested similarly by Reviewer#1, we tried to clarify our goals and connections between sections. For detailed changes, we encourage you to read the response to Reviewer#1, and we will here briefly list relevant changes:

Some paragraphs in the introduction have been reordered to follow a more progressive line of logical reasoning. The justification for exploring nitrogen isotopes is also given in the introduction.

Section 4.2 was reorganized with a general description of transects (4.2.1), description of Rayleigh distillation and isotopic enrichment (4.2.2) and validation of interpretations supported by modelling (4.2.3). With this changes, we hope that each section appears as a step progressing towards better understanding controls on nitrate uptake. The objective of each subsection has been introduced at the beginning of Sect. 4.2:

(added) *"While nitrate concentration does not appear clearly linked to sea ice, investigating the isotopic signature in nitrate may reveal further constraints on nitrate uptake. Additionally, how the nitrogen isotopes relate to nitrate concentration and other environmental processes is necessary information to be able to use isotopes as a tracer of these processes. In this section, we focus on the DP and BS transects (location given in Fig. 1), with measurements of nitrate concentration and $\delta^{15}N$. We verify that the relationship between nitrate concentration and isotopes follows a Rayleigh distillation in this region (Sect. 4.2.2). We then validate our interpretations against an isotope-enabled nitrogen cycle model forced by environmental parameters taken from three characteristic sub-regions (Sect. 4.2.3)."*

Additionally, a thorough comparison with other sectors and studies is lacking. Specifically, are these findings representative of the seasonal ice zone as a whole? Addressing these aspects would enhance the manuscript's clarity and contextual relevance.

We justified the original focus on this region in the Section 2 with the following paragraph:

(added) "*While some studies have described the relationship between sea ice and phytoplankton development in other regions of the SO (e.g. von Berg et al., 2020; Briggs et al., 2018; Taylor et al., 2013), we focus here on the region around the northern tip of the Antarctic Peninsula, from DP to WS with a particular focus on the BS. West Antarctic Peninsula and BS are coastal regions with non-depleted surface iron (Jiang et al., 2019), meaning that phytoplankton growth is more susceptible to be limited by light, controlled by shading and stratification processes related to ice cover and its melt. Moreover, the West Antarctic Peninsula is the region with the fastest decreasing trend of ice concentration (Jones*

*et al., 2016), raising the question of how productivity will change with the decrease in the ice cover."*

For comparison with other sectors, we argue that the findings are changing by region, and prefer describing only the ocean around the Antarctic Peninsula for the reasons given above. For comparison with other sectors of the Southern Ocean, we prefer citing similar work that actually compared the different regions:

(added at the end of Sect. 4.1) *"Regional differences in productivity response to MLD were also found West of the AP (Vernet et al., 2008), and Chlorophyll a concentration in the BS do not appear to correlate with MLD either (Romanova et al., 2021). Given the spatial variability within the study region, it may be hard to infer the general influence of sea ice on productivity for other regions of the SO, which have different limitations. A regional comparison of sea ice extent and chlorophyll a suggests that temperature and nutricline are important drivers in spatial discrepancies (Behera et al., 2020)."*

**Minor comments:**

Line 58: Such an approach has been described well before Moreau et al. (2020), e.g., Pondaven et al. (2000; doi:10.1038/35012046) or Nelson et al. (2002, doi:10.1016/S0967-0645(02)00005-X).

The suggested articles are now cited at the end of the corresponding sentence.

Lines 138-144: You can briefly mention (1-3 sentences) that you measured nitrate+nitrite $\delta^{15}N$ instead of nitrate-only $\delta^{15}N$. This approach is entirely appropriate for evaluating nitrate assimilation in the Antarctic Zone, given the observed differences between nitrate+nitrite $\delta^{15}N$ and nitrate-only $\delta^{15}N$, which are attributed to the interconversion between nitrate and nitrite. (e.g., Kemeny et al., 2016, https://doi.org/10.1002/2015GB005350; Fripiat et al., 2019, https://doi.org/10.1016/j.gca.2018.12.003).

This is indeed a notable point. The samples described here have been analyzed with the denitrifier method, which transforms both nitrite and nitrate. We added in the methods:

*"The denitrifier method analyzes the $\delta^{15}N$ of both nitrate and nitrite. Nitrite can have substantially lower $\delta^{15}N$ due to nitrate-nitrite equilibrium isotope effect (Kemeny et al., 2016). Given this interconversion between nitrate and nitrite N, the isotopic effect of nitrate assimilation affects the $\delta^{15}N$ of both nitrate and nitrite, hence it is more accurate to consider the $\delta^{15}N$ of the sum of nitrate and nitrite when evaluating the assimilation isotope effect (Fripiat et al., 2019). Acidified water samples might have lost most of their nitrite due to volatility of nitric acid, which would bias the $\delta^{15}N$ values towards the $\delta^{15}N$ of nitrate (Fripiat et al., 2019). However, the KARP-20 samples were not acidified prior to isotopic analysis and thus likely retained both nitrate and nitrite."*

Line 188: A reference such as Spira et al. (2024, https://doi.org/10.1029/2024JC021017) may be more appropriate to describe the winter water.

We changed the reference of Spira et al. (2024), and added reference to Moreau et al. (2020) for nitrate in the winter water.

Line 234: Are you confident that a four-year spin-up period is sufficient? Have you ensured that a steady state has been reached? Kemeny et al. (2016, https://doi.org/10.1029/2018PA003478) employed a spin-up phase lasting over a hundred years with a similar model.

We compared the results after 4 years and after 100 years (figure appended to this reply). Neither the nitrate concentration nor the isotope ratio in the upper layer showed any particular difference between 4 years (blue) and 100 years (gray). The maximum difference was 0.2 µM in concentration and 0.02‰ in isotope ratio. Consequently, we think this model has reached a steady state after 4 years. Differences in complexity with the model in Kemeny et al. (2016) might explain why the model in the present study reaches a stable state much faster.

Line 279: Here is an example where a more comprehensive analysis could have been conducted, such as a circumpolar-scale comparison. For instance, by comparing satellite estimates of primary production with the timing of sea-ice retreat.

We try to study the impact of sea-ice productivity in this region in order to interpret the nitrate concentration and isotopic data described in the second part of the article, and thus try to remain specific to this region. As noted earlier, we discussed the spatial relevance of these findings at the end of Section 4.1 and argue that they are not applicable for other regions. Comparing the impact of sea ice on productivity for other sectors of the Southern Ocean would be relevant for a broader audience, but would not serve the purpose of understanding the transect data. We prefer excluding this comparison in this article.

Line 290: See also the study of Savoye et al. (2004, https://doi.org/10.1029/2003GL018946) or Sambrotto and Mace (2000, https://doi.org/10.1016/S0967-0645(00)00071-0).

We developed the discussion adding the following: *"During early summer, productivity evolves from regenerated to new production and the nitrate uptake increases as any ammonium remaining from the winter is consumed first (Savoye et al., 2004)."*

Reference to Sambrotto and Mace (2000) was added for the increase of regenerated production in the later season (L291)

Lines 312-314: Did you check the effect between sea ice (area covered or std sea ice) on MLD? This could be worthed to mention.

Yes this is the Figure 5b, discussed at Lines 328-338. We corrected several sentences where "sea ice concentration" was improperly used, when we referred to the Standard Deviation of Sea Ice Concentration (abbreviated $SD_{SIC}$). We also added a reference to Figure 5 at Line 314.

Lines 316-319: While I found the metrics "std sea ice" convincing, I would also test other
metrics. Have you tried other metrics (e.g., difference in sea-ice concentration, time
difference between maximum and minimum sea-ice extent …)?

(This answer is the same as for a similar comment by Reviewer#1)

We tested other definitions, notably maximum minus minimum, and average sea ice. We
considered amplitude to be unreliable because it uses the measurement for two dates rather
than the full annual cycle, and is thus more susceptible to analytical errors or may even
especially highlight outliers that have not been eliminated during sea ice concentration data
processing. Moreover, the correlation with nitrate concentration or nitrate depletion was
lower than for standard deviation. For average sea ice concentration, it was ruled out because
it does not necessarily imply high meltwater input, especially for Weddell Sea point where
sea ice does not completely melt in summer.

We also tried a metric for melt duration (number of days spent with area ice concentration
between 85% and 15%), which did not result in any significant correlation, so we did not use
it.

Section 4.2: The authors should consider that some of the studies referenced used nitrate-only
$\delta^{15}N$ to estimate the isotope effect, while others used nitrate+nitrite $\delta^{15}N$ (see Fripiat et al.,
https://doi.org/10.1016/j.gca.2018.12.003). Given the evidence for nitrate-nitrite
interconversion in the Antarctic Zone, where this process occurs in a closed system,
nitrate+nitrite δ15N values are more representative of assimilated $\delta^{15}N$ and should be used to
estimate the isotope effect. Based on this approach, no significant difference in isotope effect
estimates was found in the Antarctic Zone, consistent with Fig. 8 (i.e., the two data points at
60 and 80 MLD are not significantly different from the one at below 50 MLD). This
discussion should be somewhere mentioned in the text.

In this section, regarding interpretation of $\delta^{15}N$ signal we referenced mostly articles for which
data was compiled and analyzed by DiFiore et al. (2010). According to Fripiat et al. (2019),
some of the samples in this compilation were processed with acidification (methods are not
always exhaustively described in the original articles, especially regarding acid treatments of
water samples, so we rely on Fripiat et al. 2019 for this information), which would have led
to loss of $^{15}N$-poor nitrite, biasing the $\delta^{15}N$ and the apparent isotopic effect of assimilation ε
towards positive $\delta^{15}N$. The latitudinal gradient in MLD and isotopic effect of assimilation ε
presented by DiFiore et al. (2010), is thus reinterpreted by Fripiat et al. (2019) as arising from
methodological differences rather than environmental gradient. Relying on the most recent
literature, it appears that our discussion regarding light limitation effect on isotopes cannot
result from interpretation of changes in the isotopic effect of assimilation ε. The
corresponding discussion will be changed to a single paragraph:

*"Nitrate assimilation has been reported to imprint a nitrogen isotope effect of about 5 ‰ in*
*the SO (DiFiore et al., 2009, 2010; Sigman et al., 1999). Some studies have attributed greater*
*isotope effect to continuous active pumping of nitrate to offset nitrate loss by diffusion*
*through cell membrane, when assimilatory nitrate reduction is slowed by light-limited*
*cellular activity (Needoba et al., 2004; Needoba and Harrison, 2004). In the SO, light*

*limitation was proposed as a cause of higher isotopic effects in waters with deeper mixed layers, with values exceeding 8 ‰ (DiFiore et al., 2010), but that finding must be reassessed given the recently recognized role for nitrate-nitrite N isotope exchange in SO waters (Fripiat et al., 2019; Kemeny et al., 2016), which can lead to overestimation of ε if the samples have lost nitrite (e.g., in the case of sample preservation by acidification). Here, we quantify the isotopic effect of nitrate uptake following the linear regression on logarithmic concentration scale method (DiFiore et al., 2009; Appendix Fig. A2), station by station to evaluate spatial changes in fractionation effect during assimilation of nitrate by phytoplankton. Consistent with Fripiat et al. (2019), we find nitrogen isotope effects ε around 5 ‰, insensitive to MLD for MLD ≤ 50 m (Fig. 8). It is possible that still deeper MLDs are associated with higher ε, but the weakness of the regressions make this uncertain (Fig. 8). In any case, lower surface nitrate $\delta^{15}N$ and higher surface nitrate concentrations around 61 °S in the LB transect are in line with environmental limitation of nitrate assimilation, that we attribute to light limitation as MLD is larger at this location"*

Other mentions of isotopic fractionation evidence for light limitation have been redacted (abstract and conclusion)

Section 4.2.2: The added-value and motivation for the model are unclear.

Being able to reproduce the observed patterns in a model may confirm that we understood correctly the mechanisms at play. We rephrased the opening sentence to state the objectives at the beginning of the section:

*"In this final section, we use a nitrogen isotope box-model (Yoshikawa et al., 2005) to reproduce the observed differences between three basins of the studied region, which have different sea ice cover duration, and compare them with KARP-20 nitrate patterns. Modelling the nitrogen cycle allows understanding of the seasonal evolution of nitrate concentrations, whereas observations are usually made in a short time frame."*

The discussion around dual blooming (L465-472) was judged less relevant and removed to lighten this section.

We added a short conclusion for the model:

*"Overall, the model confirms the observations that sea ice seasonality is not the principal control on productivity and surface nitrate drawdown, which are rather tied to summer MLD. The $\delta^{15}N$ of nitrate increases with surface nitrate drawdown, and both vary jointly with net productivity and particulate export."*